# Complexity Bounds for Dirichlet Process Slice Samplers

**Beatrice Franzolini** [1] [*]   **Francesco Gaffi** [2] [*]

## Abstract

Slice sampling is a standard Monte Carlo technique for Dirichlet process (DP)–based models, widely used in posterior simulation. However, formal assessments of the scalability of posterior slice samplers have remained largely unexplored, primarily because the computational cost of a slice-sampling iteration is random and potentially unbounded. In this work, we obtain high-probability bounds on the computational complexity of DP slice samplers. Our main results show that, uniformly across posterior cluster-growth regimes, the overhead induced by slice variables, relative to the number of clusters supported by the posterior, is $O_{\mathbb{P}}(\log n)$. As a consequence, even in worst-case configurations, super-linear blow-ups in per-iteration computational cost occur with vanishing probability. Our analysis applies broadly to DP–based models without any likelihood-specific assumptions, still providing complexity guarantees for posterior sampling on arbitrary datasets. These results establish a theoretical foundation for assessing the practical scalability of slice sampling in DP-based models.

## 1. Introduction

Dirichlet process (DP)–based models are widely used to define flexible latent-component structures in problems where the number of clusters, groups, or latent effects is unknown. Canonical examples include mixture models for density estimation and clustering (Neal, 2000), stochastic block models for network data (Kemp et al., 2006; Xu et al., 2006), regression and random-effects models (Wade & Inácio, 2025), and species sampling problems (Balocchi et al., 2026). Beyond classical applications, DP-based constructions continue to be an active modeling tool in novel machine learning

methodologies and applications, including domain adaptation (Ling et al., 2024), open-set classification (Snell et al., 2024), covariate-informed clustering (Chakrabarti et al., 2025), and online anomaly detection (Mei & Yan, 2026). In all these settings, inference involves exploring distributions over partitions whose complexity grows with the sample size, offering substantial modeling flexibility while posing significant computational challenges.

Markov chain Monte Carlo (MCMC) algorithms for DP-based models broadly fall into two main classes: marginal and conditional methods. Marginal algorithms integrate out the random probability measure, yielding exact inference for the induced partition structure, but typically rely on sequential, one-at-a-time updates and nontrivial book-keeping (Neal, 2000). While often effective for simple mixture models and small sample sizes, these features become increasingly restrictive in large datasets and in models with complex likelihoods, such as regression settings with covariate-dependent structure (see, for instance, Dunson & Park, 2008) or relational data with non-exchangeable observations (see, for instance, Cai et al., 2016; Zhou et al., 2024). Conditional algorithms instead retain the random probability measure explicitly, commonly via the stick-breaking representation of the DP (Sethuraman, 1994). This enables joint updates of latent variables and often leads to simpler implementations. A widely used approach in this class is truncation of the stick-breaking representation at a fixed level, resulting in blocked Gibbs samplers that are computationally efficient and, for typical likelihood specifications, perform joint updates (Ishwaran & James, 2001). However, truncation fundamentally alters the model: unless the truncation level grows with the sample size, truncated algorithms introduce a hard-threshold non-vanishing bias in the posterior distribution of the partition by assigning zero probability to configurations with more clusters than the truncation allows. Although average-case error bounds under the generative model are available (Ishwaran & James, 2002; Ishwaran & Zarepour, 2002; Campbell et al., 2019; Li & Campbell, 2021), they do not provide guarantees for a fixed observed dataset.

Slice sampling offers a principled alternative that avoids both marginalization and truncation while preserving convergence to the exact posterior (Neal, 2003; Walker, 2007; Ge et al., 2015). By introducing auxiliary slice variables,

[*]Equal contribution  [1]Department of Mathematics, King's College London, UK  [2]Department of Economics, University of Bergamo, Italy. Correspondence to: Beatrice Franzolini <beatrice.franzolini@kcl.ac.uk>.

*Proceedings of the 43rd International Conference on Machine Learning*, Seoul, South Korea. PMLR 306, 2026. Copyright 2026 by the author(s).

slice samplers restrict attention to a finite subset of components at each MCMC iteration without imposing a fixed truncation level. Originally developed for DP mixture models in Walker (2007), slice sampling extends directly to a broad class of DP-based models, generalizations of DPs, and discrete random measures whenever a stick-breaking or weight-based representation is available (Kalli et al., 2011; Zhu et al., 2020). In practice, slice samplers combine the advantages of marginal and blocked methods: they target the true posterior distribution, typically allow joint updates, and require minimal bookkeeping.

Despite these advantages, slice samplers for the DP (as defined for instance in Walker, 2007; Kalli et al., 2011; Ge et al., 2015) suffer from a fundamental theoretical limitation: their per-iteration computational cost is unbounded. The number of components that must be instantiated at a given iteration depends on the minimum slice variable, which can be arbitrarily close to zero with positive probability, forcing the number of steps in the algorithm to be arbitrarily large. Formally, let $(X_s)_{s \geq 1}$ denote the Markov chain induced by the slice sampler, $C(X_s)$ the computational cost of iteration $s$ (in some unit of measure), and $\pi$ the stationary distribution. For any finite threshold $C < \infty$, define the high-cost set $A_C := \{x : C(x) > C\}$. At stationarity, the marginal probability $\mathbb{P}_\pi(C(X_s) > C) =: \pi(A_C)$ is constant across iterations. Defining the hitting time $\tau_C = \inf\{s \geq 1 : X_s \in A_C\}$, since $\pi(A_C) > 0$, $\tau_C < \infty$ almost surely and

$$\mathbb{P}_\pi\big(\exists \, s \leq T : C(X_s) > C\big) = \mathbb{P}_\pi(\tau_C \leq T) \xrightarrow[T \to \infty]{} 1.$$

Consequently, no deterministic upper bound on the computational cost of DP slice samplers exists. This lack of formal complexity guarantees has made it difficult to assess the scalability of slice-based inference.

In this work, we provide a formal analysis of the computational complexity of slice sampling for a broad class of DP-based models. Rather than seeking deterministic worst-case bounds, which are unattainable for slice samplers, we adopt a high-probability perspective and derive probabilistic bounds on the number of components instantiated at each iteration. Our main results show that, with arbitrarily high probability, the computational overhead introduced by the slice variables over the number of clusters supported by the posterior grows at most logarithmically with the sample size. Thus, even in unfavorable configurations, super-linear blow-ups in per-iteration cost occur with vanishing probability. Our analysis applies broadly to general DP-based models without relying on model-specific likelihood assumptions, while providing guarantees for any input data. By establishing explicit high-probability complexity guarantees, this work provides a theoretical foundation for the practical scalability of slice-based inference in DP-based models.

## 2. Preliminaries on Dirichlet Process and Slice Samplers

We define a general class of models based on the DP as follows.

**Definition 2.1** (DP–based generative model). Let $\mathbf{z}_n = (z_1, \ldots, z_n)$. We say that a random object $Y$ is generated from a DP–based model with sample size $n$ if

$$Y \mid \eta, \mathbf{z}_n \sim \mathcal{L}(\mathbf{z}_n, \eta),$$
$$z_i \mid G \overset{\text{iid}}{\sim} G, \qquad i \in [n],$$
$$G \sim \text{DP}(\alpha, P_0),$$

where $\mathcal{L}$ denotes a likelihood depending on the latent variables $\mathbf{z}_n$ and parameters $\eta$, $[n] = \{1, \ldots, n\}$, and $\text{DP}(\alpha, P_0)$ is the law of a DP (Ferguson, 1973) with concentration parameter $\alpha > 0$ and non-atomic base measure $P_0$.

For different choices of the likelihood $\mathcal{L}$, Definition 2.1 encompasses a wide range of models, including species sampling models, DP mixture models, DP stochastic block models, and related constructions.

A fundamental challenge in sampling (both *a priori* and *a posteriori*) from models in Definition 2.1 arises from the infinite-dimensional nature of the random probability measure $G$. Almost surely, $G$ admits the stick-breaking representation (Sethuraman, 1994)

$$G(\mathrm{d}x) = \sum_{k=1}^{\infty} \pi_k \delta_{\phi_k}(\mathrm{d}x), \tag{1}$$

where the atoms $(\phi_k)_{k \geq 1}$ and the weights $(\pi_k)_{k \geq 1}$ are independent, with

$$\phi_k \overset{\text{iid}}{\sim} P_0, \quad \pi_k = V_k \prod_{\ell=1}^{k-1}(1 - V_\ell), \quad V_k \overset{\text{iid}}{\sim} \text{Beta}(1, \alpha).$$

This representation highlights the intrinsic infinite-dimensional structure of $G$, which is the main source of both the flexibility of DP–based models and the computational challenges associated with performing inference under them. The latent variables $\mathbf{z}_n$ in Definition 2.1 naturally induce a random partition of the index set $[n]$. Specifically, define an equivalence relation on $[n]$ by declaring $i \sim j$ if and only if $z_i = z_j$. The resulting equivalence classes correspond to clusters of latent variables sharing the same value and define a partition $\rho_n$ of $[n]$. Under the DP prior law on $G$, this random partition is exchangeable, and its *a priori* distribution depends solely on $\alpha$ (Pitman, 1996). In the following, we encode such a partition also with a *cluster-label vector* $\mathbf{c}_n = (c_1, \ldots, c_n)$, such that $c_i \in [n]$ and $c_i = c$ means that $i$ belongs to cluster $c$ (in some order, *e.g.*, order of appearance) according to $\rho_n$.

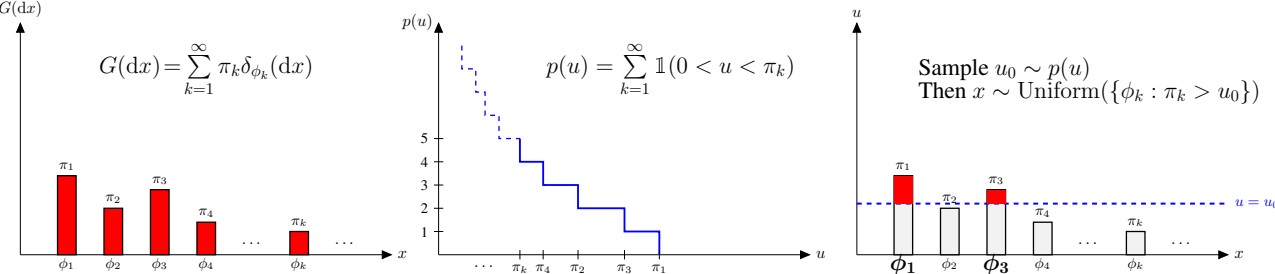

*Figure 1.* Distributions involved in the slice sampler. Sampling $x \sim G(\mathrm{d}x)$ is equivalent to sampling $u \sim p(u)$ and $x \mid u \sim p(x \mid u)$. *Left panel:* A realization of a discrete probability measure $G(\mathrm{d}x) = \sum_{k=1}^{\infty} \pi_k \delta_{\phi_k}(\mathrm{d}x)$, represented as point masses located at atoms $\phi_k$ with weights $\pi_k$. Sampling $x \sim G$ corresponds to selecting one atom $\phi_k$ with probability $\pi_k$. *Center panel:* The marginal distribution of the auxiliary slice variable $u$, given by $p(u) = \sum_{k \geq 1} \mathbb{1}(0 < u < \pi_k)$, which is a decreasing step function supported on $(0, \max_k \pi_k)$. *Right panel:* Slice-sampling mechanism. First, a slice value $u_0$ is drawn from $p(u)$ (horizontal dashed line). Conditionally on $u_0$, the variable $x$ is sampled uniformly from the finite set $\{\phi_k : \pi_k > u_0\}$, corresponding to atoms whose weights exceed the slice level.

Slice sampling for DP–based models introduces auxiliary variables to restrict inference to a finite, data-dependent subset of latent components while preserving exactness of the target distribution (*i.e.*, the posterior of $\rho_n$). The original formulation of slice sampling procedures to sample $x$ from a distribution with density $f$ on $\mathbb{X} \subset \mathbb{R}^m$ (Neal, 2003) introduces a latent slice variable $u$ with joint density $p(x, u) = \mathbb{1}(0 < u < f(x))$ that leaves the marginal distribution of $x$ unchanged. Conditional on $u$, sampling $x$ reduces to sampling uniformly over the slice set $\{x : f(x) > u\}$. This construction specializes naturally to discrete distributions (Walker, 2007; Kalli et al., 2011; Ge et al., 2015). In particular, for a DP realization as in (1), conditionally on the latent slice variable $u$, a finite active set is defined as $A(u) = \{k : \pi_k > u\}$, and $x$ is sampled uniformly from $A(u)$. Figure 1 exemplifies the distributions and the sampling mechanism that define a slice sampler for a discrete distribution $G$. Algorithms 1 and 2 contain the slice sampling algorithms to sample, respectively, from the generative model and from the posterior distribution for any model following Definition 2.1. In contrast with the procedure proposed in Walker (2007) and Kalli et al. (2011), Algorithm 2 relies on the posterior representation of the DP described in Pitman (1996) and employed in the improved slice sampler of Ge et al. (2015), which allows one to recover exchangeability of components, improving the mixing of the slice sampler and leading to the update of the weights of the allocated components according to $(\pi_1, \ldots, \pi_H, \pi^\star) \sim \mathrm{Dirichlet}(n_1, \ldots, n_H, \alpha)$. See Section 3 for a more detailed account of the implications of this choice.

While all steps involve sampling from relatively simple distributions, the core of the algorithm arguably lies in determining how many weights and atoms must be instantiated. To guarantee that the set $A(u_i)$ can be identified exactly for each $i$, a sufficient number of components must be instantiated so that all weights satisfying $\pi_k > u_i$ are available.

---

**Algorithm 1** Slice-Sampler a priori (generative mechanism)

**Require:** Hyperparameters $\alpha, P_0, \eta$, num. of iterations $T$
1: Initialize allocations $c_i$ for $i \in [n]$
2: **for** $t = 1, \ldots, T$ **do**
3:      $H \leftarrow$ number of clusters
4:      Remap $(c_i)_{i=1}^n$ into $[H]^n$
5:      $\pi^\star \leftarrow 1$
6:      **for** $h = 1, \ldots, H$ **do**
7:          Sample $\phi_h \sim P_0$
8:          Sample $v_h \sim \mathrm{Beta}(1, \alpha)$
9:          $\pi_h \leftarrow v_h \times \pi^\star$
10:         $\pi^\star \leftarrow \pi^\star - \pi_h$
11:      **end for**
12:      **for** $i = 1$ **to** $n$ **do**
13:          Sample $u_i \sim \mathrm{Uniform}(0, \pi_{c_i})$
14:      **end for**
15:      $u^* \leftarrow \min_i u_i$, $K \leftarrow H$
16:      **while** $\pi^\star > u^*$ **do**
17:          $K \leftarrow K + 1$
18:          Sample $\phi_K \sim P_0$
19:          Sample $v_K \sim \mathrm{Beta}(1, \alpha)$
20:          $\pi_K \leftarrow v_K \times \pi^\star$
21:          $\pi^\star \leftarrow \pi^\star - \pi_K$
22:      **end while**
23:      **for** $i = 1$ **to** $n$ **do**
24:          $A_i \leftarrow \{k \in [K] : \pi_k > u_i\}$
25:          Sample $c_i \sim \mathrm{Uniform}(A_i)$
26:          $z_i \leftarrow \phi_{c_i}$
27:      **end for**
28:      Sample $Y \sim \mathcal{L}(z_n, \eta)$
29: **end for**

---

In Neal's original slice sampler (Neal, 2003), the slice set is located via generic expansion procedures that are well-suited to continuous targets but possibly inefficient when $G$ is discrete. The practical difficulty of controlling the

**Algorithm 2** Slice-Sampler a posteriori

---

**Require:** Hyperparameters $\alpha$, $P_0$, $\eta$, num. of iterations $T$
1:  Initialize allocations $c_i$ for $i \in [n]$
2:  **for** $t = 1, \ldots, T$ **do**
3:     $H \leftarrow$ number of clusters
4:     Remap $(c_i)_{i=1}^n$ into an element in $[H]^n$
5:     $n_h \leftarrow \sum_i \mathbb{1}(c_i = h)$
6:     $(\pi_1, \ldots, \pi_H, \pi^\star) \sim \text{Dirichlet}(n_1, \ldots, n_H, \alpha)$
7:     **for** $h = 1$ **to** $H$ **do**
8:       Sample $\phi_h$ from its full conditional
9:     **end for**
10:    **for** $i = 1$ **to** $n$ **do**
11:      Sample $u_i \sim \text{Uniform}(0, \pi_{c_i})$
12:    **end for**
13:    $u^* \leftarrow \min_i u_i$,  $K \leftarrow H$
14:    **while** $\pi^\star > u^*$ **do**
15:      $K \leftarrow K + 1$
16:      Sample $\phi_K \sim P_0$
17:      Sample $v_K \sim \text{Beta}(1, \alpha)$
18:      $\pi_K \leftarrow v_K \times \pi^\star$
19:      $\pi^\star \leftarrow \pi^\star - \pi_K$
20:    **end while**
21:    **for** $i = 1$ **to** $n$ **do**
22:      $A_i \leftarrow \{ k \in [K] : \pi_k > u_i \}$
23:      Sample $c_i \sim \text{Cat}\big(w_k \propto \text{Lik}(Y; z_i = \phi_k), k \in A_i\big)^\star$
24:      $z_i \leftarrow \phi_{c_i}$
25:    **end for**
26:  **end for**

---

$\star$ $\text{Lik}(Y; z_i = \phi_k)$ denotes the likelihood evaluated at $z_i = \phi_k$ and current state of the other parameters. Typically, this step does not require full evaluation of the likelihood (see Section 4).

---

number of instantiated components under this approach for discrete distributions is evident, for instance, in the exact slice sampler for hierarchical DP proposed by Amini et al. (2019), which relies on additional mechanisms to manage the growth of the active set.

In this regard, Walker (2007) observes that a finite number $K$ of components sufficient to allocate the latent variables in $\boldsymbol{z}$ at each iteration is

$$K = \min\left\{ k \in \mathbb{N} : \sum_{h=k+1}^{\infty} \pi_h < u_{\min} \right\} \qquad (2)$$

with $u_{\min} := \min_i u_i$. By this definition, $K$ is finite and any weight beyond the $K$-th component necessarily satisfies $\pi_j \leq u_i$ for every $i$ and $j > K$. These components can be safely ignored at the current iteration for sampling $\boldsymbol{z}$. To implement this approach, one may dynamically determine the truncation level $K$ during the sampling of the stick-breaking weights by checking the condition $\sum_{k=1}^{K} \pi_k > 1 - u_{\min}$ and stopping as soon as it is met, as detailed in steps 16–22 of Algorithm 1 and steps 14–20 of Algorithm 2. In this

way, the algorithm avoids accept-reject steps, retrospective sampling (Papaspiliopoulos & Roberts, 2008), or any approximated truncated representation.

However, $u_{\min}$ can be arbitrarily close to zero with positive probability at each iteration, and therefore, the condition

$$\sum_{j=1}^{K} \pi_j > 1 - u_{\min}$$

may force $K$ to be arbitrarily large, so that the sampler must update an unbounded number of components (an effect that becomes more pronounced as the tails of the Dirichlet process become heavier). Therefore, it is important to assess the probability of actually exceeding a specific computational threshold at each iteration and how such probability varies with the sample size $n$ growing. In Section 3, we derive *a posteriori* high-probability bounds on computational cost, valid in any cluster-growth regimes, providing guarantees that hold a posteriori for any dataset. All the proofs are given in Appendix A.

## 3. High-probability Bounds for Slice Sampler Complexity

The computational cost of Algorithm 2 depends on the truncation level $K$ instantiated at each iteration. In particular, computational complexity is driven by the final **for**-loop updating the allocations. It requires, for each observation $i \in [n]$, evaluating the likelihood for all components $k \in A_i \subseteq [K]$. In the worst case, this step entails up to $n \times K$ likelihood evaluations per iteration. Consequently, while the slice sampler avoids fixed truncation, its practical scalability hinges on probabilistic control of the magnitude of $K$, which directly governs the dominant per-iteration computational cost. In the following, we show that the performance of the slice sampler for DP–based models is safe-guarded by high-probability bounds. To derive the result, the main quantity under study is the dynamically-determined truncation level $K$ and its behavior as $n \to \infty$, therefore from now on we replace the notation $K$ with $K_n$ to highlight dependence on the sample size.

The truncation level $K_n$, as defined by Walker (2007) and reported in equation (2), is the smallest index such that the remaining stick-breaking mass falls below the minimum slice variable $u_{\min}$. When the stick-breaking weights entering this definition are sampled from the Dirichlet process prior and one conditions on a fixed value of $u_{\min}$, the distribution of $K_n$ admits an explicit characterization: by a classical result of Muliere & Tardella (1998), one has $K_n - 1 \mid u_{\min} \sim \text{Poisson}(\alpha \log(1/u_{\min}))$. This result serves as a useful reference point for understanding how the slice level controls the number of instantiated components under prior sampling. However, its scope is inherently lim-

ited: it is based on prior sampling for the weights, does not account for the configuration of the allocated components, and, crucially, does not describe how $u_{\min}$ itself behaves as the sample size grows.

In practical implementations of slice-based Gibbs samplers, both *a priori* and *a posteriori*, the truncation level $K_n$ is subject to an additional structural constraint: it must always be at least as large as the maximum label appearing in the current cluster allocation. Otherwise, the slice variables $u_i$ cannot be sampled (see line 13 of Algorithm 1 and line 11 of Algorithm 2). The improved slice sampler of Ge et al. (2015) exploits the exchangeability of the posterior representation of the DP described in Pitman (1996) to enforce this constraint optimally, by remapping cluster labels so that the maximum label coincides with the number of occupied clusters. This relabeling step leads to a more efficient Gibbs sampler without altering the target posterior distribution. Differently from an approach relying solely on a fixed stick-breaking ordering, the improved slice sampler of Ge et al. (2015) avoids the forced instantiation and sampling of weights corresponding to empty components before entering the **while**-loop. Denoting by $H_n$ the number of clusters at the current iteration, the effective truncation level used by the algorithm can therefore be written as

$$K_n = \min \left\{ k \geq H_n : \sum_{h=k+1}^{\infty} \pi_h < u_{\min} \right\}. \quad (3)$$

In posterior inference, while the formal definition of $K_n$ in (3) remains unchanged, the probabilistic structure governing it differs substantially from the prior-based setting. Both the weights and the slice variables are sampled from their posterior distributions, conditionally on the currently visited partition configuration, which itself depends on the observed data. As a consequence, the tail mass appearing in the definition of $K_n$ is governed by posterior rather than prior-distributed stick-breaking factors, and the minimum slice variable $u_{\min}$ has a data-dependent distribution.

Understanding the computational complexity of posterior slice sampling therefore requires controlling the joint behavior of the sampled weights and the minimum slice variable $u_{\min}$ as the sample size increases, without relying on prior laws or conditioning on a fixed value of $u_{\min}$. We start investigating the law of the minimum slicing variable conditioning on the partition configuration visited by the chain and marginalizing the weights. The next Proposition highlights how merging two clusters always reduces the conditional probability of observing minimum slicing variables below any given threshold.

**Proposition 3.1** (Merging two clusters increases the survival probability of $u_{\min}$). *For any $\rho_n$ partition of $[n]$ with cluster sizes $(n_1, \ldots, n_H)$, $\sum_{h=1}^{H} n_h = n$, let*

$(\pi_1, \ldots, \pi_H, \pi^\star) \mid \rho_n \sim \text{Dirichlet}(n_1, \ldots, n_H, \alpha)$, *let $u_i \mid \pi_{c_i} \sim \text{Uniform}(0, \pi_{c_i})$ independently, and $u_{\min} = \min_{1 \leq i \leq n} u_i$.*

*If $\rho_n^{(r \oplus s)}$ is the partition obtained from $\rho_n$ by merging two (distinct) clusters $r$ and $s$ into a single cluster of size $n_r + n_s$ and leaving the other clusters unchanged, then, for any $x \in (0, 1)$*

$$\mathbb{P}\left( u_{\min} \leq x \mid \rho_n^{(r \oplus s)} \right) \leq \mathbb{P}\left( u_{\min} \leq x \mid \rho_n \right).$$

Intuitively, Proposition 3.1 reflects the fact that splitting a cluster into two divides the associated weight, yielding smaller weights and hence narrower supports for the corresponding slice variables. This reduces the probability that slice variables exceed a given threshold. As a direct consequence of Proposition 3.1, we have that the partition with $n$ clusters of size 1 induces the lowest survival probability for $u_{\min}$.

**Corollary 3.2** (Singleton partition yields the lowest survival probability of $u_{\min}$). *Let $\rho_n^{\text{sing}}$ denote the singleton partition of $[n]$, i.e., the partition with $n$ clusters of size 1. Then, for every $x \in (0, 1)$ and every partition $\rho_n$,*

$$\mathbb{P}\left( u_{\min} > x \mid \rho_n^{\text{sing}} \right) \leq \mathbb{P}\left( u_{\min} > x \mid \rho_n \right).$$

Once it is formally established that the singleton partition represents the worst-case scenario in terms of controlling the mass around zero of the minimum slice variable, we can state our main result on the tail of $K_n$, which holds uniformly over all partitions visited by the posterior algorithm.

**Theorem 3.3** (High-probability bound on dynamic truncation level). *Let $K_n$ and $H_n$ be respectively the truncation level in (3) and the number of occupied clusters at a given iteration of Algorithm 2 when run for $n$ input data. Then, for every $\delta \in (0, 1)$ there exist constants $B_\alpha^{(1)}$, $B_\alpha^{(2)}$ (independent of $\delta$ and $n$) such that for any $n \geq 2$*

$$\mathbb{P}(K_n - H_n \leq C_\delta \log n \mid \rho_n) \geq 1 - \delta, \qquad \forall \rho_n$$

*with $C_\delta = B_\alpha^{(1)} + B_\alpha^{(2)} \log(1/\delta)$, and $B_\alpha^{(1)}$, $B_\alpha^{(2)} \asymp \alpha$.*

*In particular, $K_n - H_n = O_\mathbb{P}(\log n)$ and, in the worst-case scenario of $H_n = O(n)$, $K_n = O_\mathbb{P}(n)$.*

The strength of providing: (i) an explicit constant $C_\delta$ with logarithmic growth in $\delta$ and (ii) uniformity in $n$ for the high-probability bound is exemplified in the following Corollary, which establishes exponential tails and, consequently, uniformly bounded moments for the slice overhead.

**Corollary 3.4** (Exponential tails of the slice overhead). *Let $(K_n, H_n)$ be defined as in Theorem 3.3. There exist constants $B_\alpha^{(1)}, B_\alpha^{(2)} > 0$ such that for all $n \geq 2$ and all $t \geq 0$,*

$$\mathbb{P}\left( \frac{K_n - H_n}{\log n} > B_\alpha^{(1)} + B_\alpha^{(2)} t \mid \rho_n \right) \leq e^{-t}, \qquad \forall \rho_n.$$

|  | CRP | BGS–$L$ | BGS–$n$ | Slice |
|---|---|---|---|---|
| Scalability by posterior cluster growth | | | | |
| $H_n = O(n)$ | $O(n^2)$ | $\Theta(n)$ | $\Theta(n^2)$ | $O_{\mathbb{P}}(n^2)$ |
| $H_n = O(\log n)$ | $O(n \log n)$ | $\Theta(n)$ | $\Theta(n^2)$ | $O_{\mathbb{P}}(n \log n)$ |
| $H_n = O(1)$ | $O(n)$ | $\Theta(n)$ | $\Theta(n^2)$ | $O_{\mathbb{P}}(n \log n)$ |
| Exact posterior partition target | ✓ | ✗ | ✗ | ✓ |
| No hard-threshold bias | ✓ | ✗ | ✓ | ✓ |
| No bookkeeping | ✗ | ✓ | ✓ | ✓ |
| Joint Updates | ✗ | ✓ | ✓ | ✓ |

*Table 1.* Comparison of posterior sampling algorithms for DP mixture models. The table reports per-iteration computational complexity as a function of the sample size $n$ and the number of clusters $H_n$ of the partition visited by the chain, together with key qualitative properties of each method. Since the number of clusters $H_n$ visited by the chain can grow at most linearly in $n$, the first row represents the worst-case cluster growth regime and thus provides worst-case computational guarantees.

*In particular, for any $p \geq 1$, there exists a constant $C_{p,\alpha} > 0$ such that*

$$\sup_{n \geq 2} \mathbb{E}\left[ \left( \frac{K_n - H_n}{\log n} \right)^p \,\Big|\, \rho_n \right] \leq C_{p,\alpha}, \qquad \forall \rho_n.$$

Moreover, the uniform-in-$\rho_n$ nature of the high-probability bound allows for further derivation of an almost-sure bound under infinite-data coupling, implying that super-logarithmic slice overheads cannot occur infinitely often as the sample size grows.

**Corollary 3.5** (Almost-sure control of super-logarithmic slice overheads)**.** *Let $(K_n, H_n)$ be defined as in Theorem 3.3. There exists a constant $D_\alpha > 0$ such that*

$$\mathbb{P}\left( \limsup_{n \to \infty} \left\{ K_n - H_n > D_\alpha \log \frac{1}{\delta_n} \log n \right\} \right) = 0$$

*for any summable sequence $(\delta_n)_{n \geq 1} \subset (0, 1/2)$.*

We close this section by showing that the logarithmic rate in Theorem 3.3 is tight as a uniform upper bound on the overhead. Indeed, no $o(\log n)$ rate can hold uniformly over all partitions, as shown by the next proposition.

**Proposition 3.6** (Tightness of the logarithmic slice-overhead bound)**.** *Let $(K_n, H_n)$ be defined as in Theorem 3.3. Then there exist constants $c_\alpha, \eta_\alpha > 0$ such that for all sufficiently large $n$,*

$$\mathbb{P}\left( K_n - H_n \geq c_\alpha \log n \,\big|\, \rho_n^{\mathrm{sing}} \right) \geq \eta_\alpha.$$

*Consequently, for any deterministic sequence $r_n = o(\log n)$ and any $M > 0$,*

$$\liminf_{n \to \infty} \sup_{\rho_n} \mathbb{P}\left( K_n - H_n > M r_n \,|\, \rho_n \right) > 0.$$

## 4. Comparison with Alternative Algorithms for Mixture Models

To translate the results of the previous section into explicit high-probability statements on the scalability of slice sampling for DP models, we focus, for concreteness, on DP mixtures of univariate Normals with fixed variance. Specifically, we consider models of the form

$$Y_i \mid z_i \overset{\text{ind}}{\sim} \mathcal{N}(z_i, \sigma^2), \quad z_i \mid G \overset{\text{iid}}{\sim} G, \quad G \sim \mathrm{DP}(\alpha, P_0),$$

for $i \in [n]$, where $\mathcal{N}(\mu, \sigma^2)$ denotes a Normal distribution with mean $\mu$ and variance $\sigma^2$. This setting serves as a representative working example in which the cost of likelihood evaluation at step 23 of Algorithm 2 is constant per component. However, the arguments developed below extend directly to the broader class of DP–based models described in Definition 2.1, with the understanding that model-specific likelihoods may introduce different per-evaluation computational costs as a function of $n$. Table 1 summarizes the computational scalability and qualitative properties of several MCMC posterior sampling strategies for DP mixture models, highlighting fundamental trade-offs between exactness, scalability, and implementation complexity.

Marginal samplers based on the Chinese restaurant process (CRP) representation target the exact posterior distribution over partitions and avoid truncation bias. Their dominant per-iteration cost scales as $n \times H_n$, corresponding to the reassignment of each observation across all currently occupied clusters. While this scaling captures the leading-order behavior, the associated constant depends on implementation details. In particular, when cluster-specific parameters are analytically integrated out from the full-conditionals of the latent cluster-label vector $c_n$, each reassignment requires evaluating a predictive likelihood that typically involves an integral with respect to the base measure of the DP. Moreover, bookkeeping operations needed to maintain sufficient

statistics and cluster counts, while not altering the $n \times H_n$ scaling, can substantially increase the constant factor and limit practical scalability to large datasets, while assignments of observations to different clusters, *i.e.*, sampling the elements in $c_n$, can only be performed one-at-a-time.

Blocked Gibbs samplers with fixed truncation levels (BGS–$L$) enable simpler implementations and joint updates, *i.e.*, the assignments to different clusters can be performed in parallel, but at the cost of introducing a hard truncation. This is a model misspecification, not just an approximation error, and leads to a non-vanishing bias. In this regard, (Ishwaran & James, 2002) establishes in their Theorem 1 and Corollary 1 that the marginal density error of an $L$-truncated approximation of the DP mixture of Normals is exponentially accurate for the posterior of the partition. Formally, denote by $\pi_L$ the posterior of the truncated model, and by $\pi_\infty$ and $m_\infty$ the posterior and the marginal likelihood for the DP mixture of Normals. Ishwaran & James (2002) proves that

$$\int_{\mathbb{R}^n} \sum_{\rho_n} |\pi_L(\rho_n \mid \boldsymbol{Y}) - \pi_\infty(\rho_n \mid \boldsymbol{Y})| \, m_\infty(\boldsymbol{Y}) \mathrm{d}\boldsymbol{Y}$$

$$= O(4 \, n \, e^{-(L-1)/\alpha}).$$

While this result gives a useful average-case rate under the prior predictive, it clearly does not guarantee small error for any fixed dataset (and, in particular, for the worst-case scenario) since large discrepancies may occur on regions of the data space with small $m_\infty$-mass. The truncated model simply cannot represent more than $L$ clusters: $\pi_L(\rho_n) = 0$ on all partitions with more than $L$ clusters, yet these partitions may have substantial $\pi_\infty$-mass. Since the truncation level is fixed by design, their per-iteration cost is deterministic and scales as $nL$. One way to avoid the systematic underestimation of the probability of partitions with $H_n > L$ in truncated blocked algorithms is to let the truncation level be exactly $n$ (BGS-$n$). This strategy, however, leads to a $\Theta(n^2)$ per-iteration computational cost in all scenarios, while the chain still does not target the exact posterior.

On the other hand, the scalability of slice-based samplers is effectively safeguarded by our high-probability bound: the excess number of sampled components beyond those occupied by the data grows at most logarithmically with high probability. Consequently, as summarized in Table 1, slice samplers retain exactness, easy or no bookkeeping, and joint updates while achieving favorable scalability in high-probability, especially in regimes where the number of clusters grows slowly with the sample size.

## 5. Numerical Experiments

First, in this section, we present a numerical study aimed at empirically comparing the computational performance of six posterior sampling algorithms for DP mixtures of univariate Normals. The methods considered are: the slice sampler described in Algorithm 2; a variant of the slice sampler in which the atoms $\phi_k$ are analytically marginalized out; two blocked Gibbs samplers based on finite dimensional Dirichlet distributions with truncation levels $L = 10$ and $L = n$, respectively; and two marginal samplers based on the CRP predictive scheme, one sampling the cluster allocations $c_n$ conditionally on the atoms $\phi_k$ and one integrating the atoms out analytically.

Data are generated from three equally sized clusters, with observations in each cluster drawn independently from a Normal distribution with unit variance and means equal to $-3$, $0$, and $3$. Experiments are repeated for sample sizes $n \in \{150, 300, 600, 1500, 3000, 7500, 12000\}$. For all methods, the base measure of the DP is taken to be the standard Normal distribution, and the concentration parameter $\alpha$ is assigned a $\mathrm{Gamma}(3, 3 \log n)$ prior, and, thus $\mathbb{E}[\alpha] = 1/\log(n)$ a priori. For each algorithm and sample size, we run 10,000 MCMC iterations. All chains are initialized using a $k$-means clustering solution with five clusters, obtained via the R function `kmeans`.

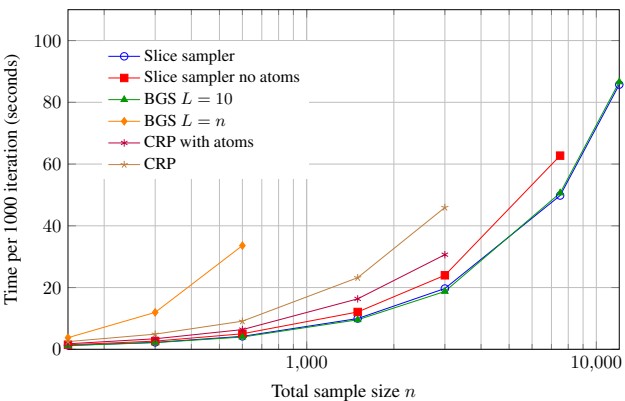

*Figure 2.* Wall-clock time per 1000 iterations in seconds computed as average over 10,000 iterations (including burn-in) as a function of input size (x-axis on log scale). Codes are in R and run on a laptop with 13th Gen Intel(R) Core(TM) i7-1370P.

All algorithms are implemented in R and are made available as supplementary material. Figures 2 and 3 report, respectively, the wall-clock time required for 1,000 iterations (in seconds) and the effective sample size (ESS) per second as functions of the input size. For some methods, results at larger sample sizes are unavailable due to computational constraints; we declare an algorithm to be infeasible at a given sample size if it requires more than one second to complete the first ten iterations on a laptop with 13th Gen Intel(R) Core(TM) i7-1370P. Marginal CRP-based samplers exhibit the steepest growth in computational time, becoming infeasible already at moderate sample sizes, while blocked Gibbs samplers display predictable scaling behavior. In

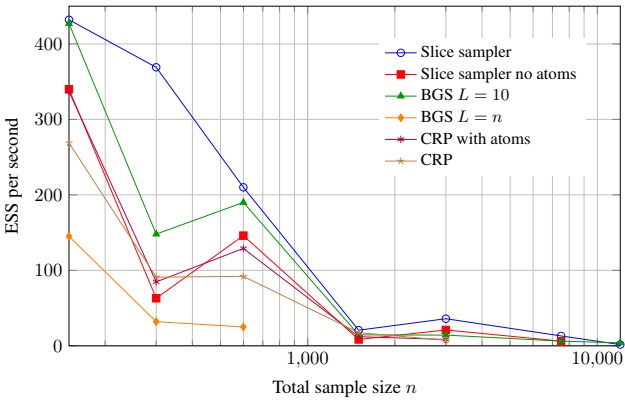

*Figure 3.* ESS (w.r.t. the log likelihood) per second computed as average over the last 5,000 iterations (excluding a burn-in of 5,000) as a function of input size (x-axis on log scale). Codes are in R and run on a laptop with 13th Gen Intel(R) Core(TM) i7-1370P.

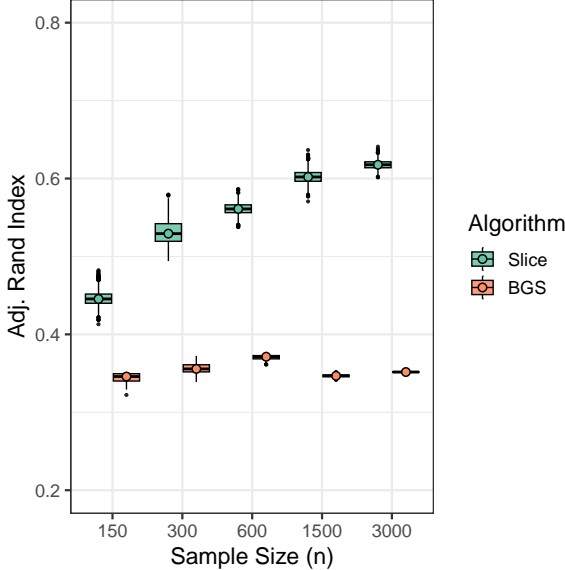

*Figure 4.* Boxplots of the adjusted Rand index between the true clustering and the partitions visited by the MCMC algorithms, for each algorithm and sample size. Results are computed after a burn-in period of 5,000 iterations and thinning every 2 iterations.

contrast, slice-based samplers achieve a more favorable trade-off: although their per-iteration cost increases with $n$, the growth is substantially slower than that of marginal samplers and comparable to that of blocked Gibbs samplers with $L = 10$ components, which, however, do not target the correct posterior. This behavior is also reflected in high ESS per second. Marginalizing the atoms within the slice sampler appears to increase the computational time per iteration but does not improve mixing sufficiently, particularly for small datasets, leading to lower ESS per second for small sample sizes compared to the original version. Overall, these empirical results align with the high-probability complexity bounds established in Section 3, supporting the scalability of slice-based approaches for posterior inference in DP–based models.

To conclude, we empirically assess the limitations of blocked Gibbs samplers compared to slice samplers with a second numerical experiment. Here, the synthetic data-generating mechanism follows a *perturbed Zipf* model, which entails a steadily increasing cluster-growth regime, up to large sample sizes, while still having finite support over the cluster labels. Specifically, for each sample size $n \in \{150, 300, 600, 1500, 3000\}$, latent cluster labels are generated independently from a discrete distribution on $\{1, \ldots, 50\}$, with the probability of cluster $c$ proportional to $c^{-0.5}$. This induces a heavy-tailed cluster-size distribution with a growing number of small but non-negligible clusters, providing a challenging setting, in particular for truncated algorithms. Conditional on the latent labels, observations are drawn independently from Normal distributions with unit variance and cluster-specific means proportional to the label index. With the exception of the data-generating process, all other settings coincide with those of the previous numerical study.

The results are reported in Figure 4. As expected, the in-

ferential performance of the blocked Gibbs sampler deteriorates rapidly as the sample size increases, with the chain exhibiting very limited mixing and remaining almost constant on a single partition. In contrast, the slice sampler consistently achieves higher and increasing adjusted Rand index values across all sample sizes, indicating good recovery of the true clustering even in this challenging scenario.

Additional results on posterior inference performance and computational aspects for both numerical studies are reported in Appendix B, together with results from analogous settings in which the concentration parameter is kept fixed.

## 6. Conclusion

Slice samplers provide an appealing strategy for posterior simulation in DP–based models, combining joint updates, minimal bookkeeping, and the avoidance of fixed truncation. At the same time, their practical scalability has long remained theoretically unclear due to the random and unbounded nature of their per-iteration computational cost. This work addresses this gap by providing the first high-probability complexity guarantees for DP slice samplers that hold *a posteriori*, uniformly over all partitions visited by the Markov chain and for any observed dataset. Our main results establish that the dynamically instantiated truncation level exceeds the number of occupied clusters by at most a logarithmic factor in the sample size, with arbitrarily high probability. As a consequence, even in worst-case cluster-growth regimes, super-linear increases in per-iteration cost

occur with vanishing probability. These guarantees formalize and explain the favorable empirical scalability of slice-based algorithms observed in practice, while preserving exact targeting of the posterior distribution over partitions.

An additional feature of our results is that all constants appearing in the high-probability bounds are explicit in the DP concentration parameter $\alpha$. This opens the door to extensions to adaptive and data-driven DP formulations, such as those considered by Tsiligkaridis & Forsythe (2015) and Ohn & Lin (2023), where $\alpha$ is learned from the data or evolves over time. More broadly, the techniques developed here suggest a promising avenue for extending high-probability complexity guarantees to slice-based algorithms for Pitman–Yor processes (see Canale et al., 2022), general species sampling processes (for instance building upon Mena et al., 2025), and structured DP-based models, including hierarchical Dirichlet processes, sticky constructions, and temporally evolving partition models. In many of these settings, the conditional formulation underlying slice samplers leads to substantially simpler full conditional distributions than those arising in marginal approaches, hence the computational gains relative to alternative inference schemes are expected to be even more pronounced.

## Software and Data

The code used in this work is publicly available at `https://github.com/beatricefranzolini/DPalg`.

## Acknowledgements

The authors thank the four anonymous reviewers for their helpful suggestions and insightful comments. We are also grateful to Antonio Canale and Ramsés H. Mena for valuable discussions on this work.

## Impact Statement

This paper studies the computational properties of slice sampling algorithms for Dirichlet process–based models and provides high-probability guarantees on their per-iteration complexity. The primary contribution is theoretical and methodological, with the goal of improving the understanding and scalability of exact Bayesian inference methods. As such, the results are broadly applicable across a range of machine learning and statistical models that rely on adaptive latent structure, but do not target a specific application domain.

We do not anticipate direct negative societal impacts arising from this work. Any downstream effects depend on how Dirichlet process–based models are applied in practice, which is beyond the scope of this paper. By providing clearer complexity guarantees, this work may contribute to more reliable and transparent use of Bayesian nonparametric methods in future machine learning research.

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

# A. Proofs

### A.1. Proof of Proposition 3.1

Fix $h \in [H]$ and let $I_h := \{i : c_i = h\}$ with $|I_h| = n_h$. Conditionally on $\pi_h$, the slice variables $(u_i)_{i \in I_h} \overset{\text{iid}}{\sim} \text{Uniform}(0, \pi_h)$. Hence

$$\mathbb{P}\left(\min_{i \in I_h} u_i > x \,\Big|\, \pi_h\right) = \mathbb{P}\left(\bigcap_{i \in I_h} \{u_i > x\} \,\Big|\, \pi_h\right) = \prod_{i \in I_h} \mathbb{P}\left(u_i > x \mid \pi_h\right).$$

For $u \sim \text{Uniform}(0, \pi_h)$ and $x \in (0, 1)$,

$$\mathbb{P}\left(u > x \mid \pi_h\right) = \max\left(1 - \frac{x}{\pi_h}, 0\right),$$

so that

$$\mathbb{P}\left(\min_{i \in I_h} u_i > x \,\Big|\, \pi_h\right) = \left[\max\left(1 - \frac{x}{\pi_h}, 0\right)\right]^{n_h} = \max\left(\left(1 - \frac{x}{\pi_h}\right)^{n_h}, 0\right).$$

Now, consider the survival probability conditional on weights. Grouping indices by cluster, we have

$$\mathbb{P}\left(u_{\min} > x \mid \pi_{1:H}, \rho_n\right) = \prod_{h=1}^{H} \mathbb{P}\left(\min_{i:c_i=h} u_i > x \,\Big|\, \pi_h\right) = \prod_{h=1}^{H} \max\left(\left(1 - \tfrac{x}{\pi_h}\right)^{n_h}, 0\right),$$

Denoting $(\cdot)_+ = \max(\cdot, 0)$ and taking expectation w.r.t. the Dirichlet posterior law for the weights, *i.e.*, $(\pi_1, \ldots, \pi_{H_n}, \pi^\star) \mid \rho_n \sim \text{Dirichlet}(n_1, \ldots, n_H, \alpha)$, yields

$$\mathbb{P}\left(u_{\min} > x \mid \rho_n\right) = \mathbb{E}\left[\prod_{h=1}^{H} \left(1 - \tfrac{x}{\pi_h}\right)_+^{n_h} \,\Big|\, \rho_n\right]. \tag{4}$$

Now merge clusters $r$ and $s$. Let $\tilde{\pi} := \pi_r + \pi_s$ and $S := \pi_r/\tilde{\pi}$. By the aggregation and neutrality properties of the Dirichlet distribution,

$$(\tilde{\pi}, \pi_{-rs}, \pi^\star) \mid \rho_n \sim \text{Dirichlet}(n_r + n_s, (n_h)_{h \neq r,s}, \alpha), \tag{5}$$

$S \mid \tilde{\pi}, \rho_n \sim \text{Beta}(n_r, n_s)$, and $S$ is independent of $(\tilde{\pi}, \pi_{-rs}, \pi^\star)$. Note that the distribution in (5) coincides with the Dirichlet posterior law for the weights when conditioning on the partition $\rho_n^{(r \oplus s)}$. Now, conditioning on $(\tilde{\pi}, \pi_{-rs}, \pi^\star)$ and integrating $S$,

$$\mathbb{P}(u_{\min} > x \mid \rho_n) = \mathbb{E}\left[\prod_{h \neq r,s} \left(1 - \tfrac{x}{\pi_h}\right)_+^{n_h} \mathbb{E}\left[\left(1 - \tfrac{x}{S\tilde{\pi}}\right)_+^{n_r} \left(1 - \tfrac{x}{(1-S)\tilde{\pi}}\right)_+^{n_s} \,\Big|\, \tilde{\pi}, \rho_n\right] \,\Big|\, \rho_n\right], \tag{6}$$

where the outer expectation is w.r.t. $(\tilde{\pi}, \pi_{-rs}, \pi^\star)$.

Under the merged partition $\rho_n^{(r \oplus s)}$, the two clusters are replaced by a single cluster of size $n_r + n_s$ with weight $\tilde{\pi}$, while all other weights have the same joint law as in the aggregated representation above, and, in particular, weights follow the distribution in (5). Hence, analogously to (4),

$$\mathbb{P}\left(u_{\min} > x \mid \rho_n^{(r \oplus s)}\right) = \mathbb{E}\left[\left\{\prod_{h \neq r,s} \left(1 - \tfrac{x}{\pi_h}\right)_+^{n_h}\right\} \left(1 - \tfrac{x}{\tilde{\pi}}\right)_+^{n_r + n_s} \,\Big|\, \rho_n^{(r \oplus s)}\right]. \tag{7}$$

It therefore suffices to show that

$$\mathbb{E}\left[\left(1 - \tfrac{x}{S\tilde{\pi}}\right)_+^{n_r} \left(1 - \tfrac{x}{(1-S)\tilde{\pi}}\right)_+^{n_s} \,\Big|\, \tilde{\pi}, \rho_n\right] \leq \left(1 - \tfrac{x}{\tilde{\pi}}\right)_+^{n_r + n_s} \qquad \text{a.s.} \tag{8}$$

Indeed, plugging (8) into (6) and comparing with (7) yields $\mathbb{P}\left(u_{\min} > x \mid \rho_n\right) \leq \mathbb{P}\left(u_{\min} > x \mid \rho_n^{(r \oplus s)}\right)$, which is equivalent to the desired CDF inequality.

To verify (8), note first that if $\tilde{\pi} \leq x$ then both sides are zero. Assume $\tilde{\pi} > x$ and define $y := x/\tilde{\pi} \in (0,1)$. The left-hand side integrand vanishes unless jointly $S > y$ and $1 - S > y$, i.e., $y \leq S \leq 1 - y$. On this event, $\frac{y}{S} \geq y$ and $\frac{y}{1-S} \geq y$, which implies

$$1 - \frac{y}{S} \leq 1 - y \qquad \text{and} \qquad 1 - \frac{y}{1-S} \leq 1 - y.$$

Therefore, for all $S \in (y, 1-y)$,

$$F(S) = \left(1 - \tfrac{y}{S}\right)^{n_r} \left(1 - \tfrac{y}{(1-S)}\right)^{n_s} \leq (1-y)^{n_r}(1-y)^{n_s} = (1-y)^{n_r + n_s},$$

and since $F(S) = 0$ for $S \notin (y, 1-y)$, the bound holds for all $S \in (0,1)$. Taking expectation with respect to $S \sim \text{Beta}(n_r, n_s)$ yields

$$\mathbb{E}\left[F(S)\right] \leq (1-y)^{n_r + n_s} = \left(1 - \tfrac{x}{\tilde{\pi}}\right)^{n_r + n_s}.$$

which is exactly (8).

## A.2. Proof of Corollary 3.2

Fix a partition $\rho_n$ with $H$ clusters. Starting from the singleton partition $\rho_n^{\text{sing}}$, one can obtain $\rho_n$ by a finite sequence of merges of two clusters: at each step, merge two blocks until exactly the blocks of $\rho_n$ are formed. Denote the resulting sequence of partitions by

$$\rho_n^{(0)} = \rho_n^{\text{sing}}, \ \rho_n^{(1)}, \ \ldots, \ \rho_n^{(m)} = \rho_n.$$

By Proposition 3.1, each merge weakly increases the survival probability of $u_{\min}$, i.e., for every $j = 1, \ldots, m$,

$$\mathbb{P}\left(u_{\min} > x \mid \rho_n^{(j-1)}\right) \ \leq \ \mathbb{P}\left(u_{\min} > x \mid \rho_n^{(j)}\right).$$

Chaining these inequalities yields

$$\mathbb{P}\left(u_{\min} > x \mid \rho_n^{\text{sing}}\right) \ \leq \ \mathbb{P}\left(u_{\min} > x \mid \rho_n\right).$$

## A.3. Proof of Theorem 3.3

First, we state and prove the following technical lemmas.

**Lemma A.1.** *Let $Z \sim \text{Poisson}(\alpha \log(1/x))$ for any $\alpha > 0$ and any $x \in (0,1)$. Then*

$$\mathbb{P}\left(Z \geq 3\alpha \log(1/x) + \log(2/\delta)\right) \leq \frac{\delta}{2}$$

*Proof of Lemma A.1.* We leverage the following Chernoff-style result for Poisson random variables: if $Z \sim \text{Poisson}(\lambda)$ then

$$\mathbb{P}\left(Z \geq \lambda + t\right) \leq \exp\left\{-\frac{t^2}{2(\lambda + t/3)}\right\}$$

for any $t > 0$. This can be easily proved, *e.g.* by applying the Bernstein inequality in Equation 2.10 of Boucheron et al. (2013) to a sequence of $n$ independent Bernoulli$(\lambda/n)$ and then passing to the limit. Hence, by taking $\lambda = \alpha \log(1/x)$, for any $t > 0$, $x \in (0,1)$, and $\alpha > 0$, we have

$$\mathbb{P}\left(Z \geq \alpha \log(1/x) + t\right) \leq \exp\left\{-\frac{t^2}{2(\alpha \log(1/x) + t/3)}\right\}. \tag{9}$$

The r.h.s. of (9) is bounded by $\delta/2$ if and only if

$$t^2 - \frac{2}{3}\log(2/\delta)t - 2\log(2/\delta)\alpha \log(1/x) \geq 0 \tag{10}$$

for any $x \in (0,1)$. The positive root of (10) is

$$
\begin{aligned}
t_+ =& \frac{1}{3} \log(2/\delta) + \frac{1}{2} \sqrt{\frac{4}{9} \log^2(2/\delta) + 8\alpha \log(2/\delta) \log(1/x)} \\
\leq& \frac{2}{3} \log(2/\delta) + \sqrt{2\log(2/\delta)\alpha \log(1/x)} \leq \log(2/\delta) + \frac{3\alpha}{2} \log(1/x)
\end{aligned}
\tag{11}
$$

where we used that $\sqrt{y_1 + y_2} \leq \sqrt{y_1} + \sqrt{y_2}$ and that $\sqrt{2y_1y_2} \leq \eta y_1/2 + y_2/\eta$ for any $\eta > 0$, and in particular, we set $\eta = 2/3$. Since the l.h.s of (10) is increasing around $t_+$, from (9) and (11) we have that

$$
\mathbb{P}\left(Z \geq 3\alpha \log(1/x) + \log(2/\delta)\right) \leq \frac{\delta}{2}
$$

for any $x \in (0,1)$. $\qquad\square$

**Lemma A.2.** *For any $n \geq 2$, $\alpha > 0$, and $\delta \in (0,1)$, if*

$$
x(n,\delta) := \frac{\delta}{4n(\alpha + n - 1)\log(2n(\alpha + n - 1)e/\delta)}
\tag{12}
$$

*then we have: $x(n,\delta) < 1$,*

$$
n(\alpha + n - 1)x(n,\delta)[1 + \log(1/x(n,\delta))] \leq \frac{\delta}{2},
\tag{13}
$$

*and*

$$
1 + 3\alpha \log(1/x(n,\delta)) + \log(2/\delta) \leq C_\delta \log n
\tag{14}
$$

*where*

$$
C_\delta = B_\alpha^{(1)} + B_\alpha^{(2)} \log \frac{1}{\delta} \quad with \quad B_\alpha^{(2)} = \frac{6\alpha + 1}{\log 2}, \quad B_\alpha^{(1)} = 12\alpha + \frac{1 + 3\alpha \log\left(8e(1+\alpha)^2\right) + \log 2}{\log 2}.
\tag{15}
$$

*Proof of Lemma A.2.* Let $r = \frac{\delta}{2n(n+\alpha-1)}$. For any $n \geq 2$, we have $0 < r < 1$, since $2n(\alpha+n-1) > 1$, hence $x(n,\delta) < 1$. Moreover

$$
\begin{aligned}
x(n,\delta)\left[1 + \log \frac{1}{x(n,\delta)}\right] =& \frac{r}{2\log(e/r)}\left[1 + \log\left(\frac{2\log(e/r)}{r}\right)\right] \\
=& \frac{r}{2\log(e/r)}\left[\log(e/r) + \log\left(2\log(e/r)\right)\right] \leq r
\end{aligned}
\tag{16}
$$

since $\log(2y) \leq y$ for any $y > 0$. Substituting back $r$, we have (13).

For (14), first observe that

$$
\log \frac{1}{x(n,\delta)} = \log(4/\delta) + \log n + \log(\alpha + n - 1) + \log\log\left(\frac{2n(\alpha + n - 1)e}{\delta}\right).
\tag{17}
$$

Since $\alpha + n - 1 \leq (1 + \alpha)n$ for $n \geq 2$, we have $\log(\alpha + n - 1) \leq \log(1 + \alpha) + \log n$, hence

$$
\log\log\left(\frac{2n(\alpha + n - 1)e}{\delta}\right) \leq \log\left(\frac{2n(\alpha + n - 1)e}{\delta}\right) \leq 2\log n + \log\frac{2e(1+\alpha)}{\delta}.
\tag{18}
$$

Combining (17) and (18) yields, for all $n \geq 2$,

$$
\log \frac{1}{x(n,\delta)} \leq 4\log n + \log\left(\frac{8e(1+\alpha)^2}{\delta^2}\right).
$$

Therefore

$$
1 + 3\alpha \log \frac{1}{x(n,\delta)} + \log\frac{2}{\delta} \leq 12\alpha \log n + \left(1 + 3\alpha \log\left(\frac{8e(1+\alpha)^2}{\delta^2}\right) + \log\frac{2}{\delta}\right) \leq C_\delta \log n
\tag{19}
$$

where $C_\delta$ is as defined in (15), and for the second inequality in (19) we used that $\log 2 \leq \log n$ to collect all the terms in one constant. $\qquad\square$

We can now prove the main result.

*Proof of Theorem 3.3.* Let $\rho_n$ be the random partition of $[n]$ induced by $(z_1, \ldots, z_n)$ at a given iteration of Algorithm 2, and let $H_n$ denote the number of clusters. Let $(c_1, \ldots, c_n) \in [n]^n$ be any cluster-label vector compatible with $\rho_n$. Conditionally on $\rho_n$, draw weights $(\pi_h)_{h \geq 1}$ from the Dirichlet process posterior (as detailed in steps 4–6 of Algorithm 2) and let

$$u_i \sim \text{Uniform}(0, \pi_{c_i}), \qquad u_{\min} = \min_{1 \leq i \leq n} u_i.$$

For each integer $k \geq 1$, define

$$R_k = \sum_{h \geq k+1} \pi_h, \text{ and } K_n = \min\{ k \geq H_n : R_k < u_{\min}\}.$$

Recalling the *a posteriori* representation of allocated components, *i.e.*, $(\pi_1, \ldots, \pi_H, \pi^\star) \sim \text{Dirichlet}(n_1, \ldots, n_H, \alpha)$, with $\alpha$ being the concentration parameter, and the stick-breaking construction of the weights $(\pi_h)_{h \geq 1}$, for any $k \geq H_n$ we have

$$R_k = \pi^\star \times \prod_{i=H_n+1}^{k} (1 - V_i)$$

where we use the convention that $\prod_{i=h+1}^{h} y_i := 1$ for any $h$ and $(y_i)_i$. The distribution of each stick-breaking weight $V_i$, for $i > H_n$, conditionally on $\rho_n$, is $V_i \sim \text{Beta}(1, \alpha)$. So that

$$K_n = \min\left\{ k \geq H_n : \pi^\star \times \prod_{i=H_n+1}^{k} (1 - V_i) < u_{\min} \right\}.$$

Let us define for any $x \in (0, 1)$

$$K_n(x) := \min\left\{ k > H_n : \pi^\star \times \prod_{i=H_n+1}^{k} (1 - V_i) < x \right\} \tag{20}$$

and

$$K_n^0(x) := \min\left\{ k > H_n : \prod_{i=H_n+1}^{k} (1 - V_i) < x \right\} \tag{21}$$

where, since the products are decreasing in $k$, both the sets in (20) and (21) are non-empty for any $x$. Similarly, since $\pi^\star < 1$ a.s., we have

$$\left\{ k > H_n : \pi^\star \times \prod_{i=H_n+1}^{k} (1 - V_i) < x \right\} \supseteq \left\{ k > H_n : \prod_{i=H_n+1}^{k} (1 - V_i) < x \right\}$$

for any $x \in (0, 1)$, taking the minimum

$$K_n(x) \leq K_n^0(x) \quad \text{a.s.} \quad \forall x \in (0, 1). \tag{22}$$

It is easy to see that $u_{\min} > x$ implies

$$\left\{ k > H_n : \pi^\star \times \prod_{i=H_n+1}^{k} (1 - V_i) < x \right\} \subseteq \left\{ k \geq H_n : \pi^\star \times \prod_{i=H_n+1}^{k} (1 - V_i) < u_{\min} \right\}.$$

Hence, using again a minimum argument, $K_n(x) \geq K_n$. By negation then, $K_n(x) < K_n$ implies $u_{\min} \leq x$, which means

$$\mathbb{P}\left(K_n(x) < K_n \mid \rho_n\right) \leq \mathbb{P}\left(u_{\min} \leq x \mid \rho_n\right) \tag{23}$$

for any $x \in (0, 1)$.

Now, for any non-negative quantity $A_{n,\delta}$, possibly dependent on $n$ and $\delta$, we have

$$
\begin{aligned}
\mathbb{P}\left(K_n > A_{n,\delta} \mid \rho_n\right) &\leq \mathbb{P}\left(K_n^0(x) > A_{n,\delta} \mid \rho_n\right) + \mathbb{P}\left(K_n > K_n^0(x) \mid \rho_n\right) \\
&\leq \mathbb{P}\left(K_n^0(x) \geq A_{n,\delta} \mid \rho_n\right) + \mathbb{P}\left(K_n > K_n(x) \mid \rho_n\right) \\
&\leq \mathbb{P}\left(K_n^0(x) \geq A_{n,\delta} \mid \rho_n\right) + \mathbb{P}\left(u_{\min} \leq x \mid \rho_n\right)
\end{aligned}
$$

where we used in the first inequality the law of total probability disintegrating with respect to the event $\{K_n^0(x) \geq K_n\}$ and its complement, in the second (22) and in the third (23). Hence, choosing $A_{n,\delta} = H_n + C_\delta \log n$, we just need to find $x = x(n,\delta)$ such that, for any $n \geq 2$

$$
\mathbb{P}\left(K_n^0(x(n,\delta)) \geq H_n + C_\delta \log n \mid \rho_n\right) \leq \frac{\delta}{2} \quad \text{and} \quad \mathbb{P}\left(u_{\min} \leq x(n,\delta) \mid \rho_n\right) \leq \frac{\delta}{2}. \tag{24}
$$

For the first inequality, we notice, as in Muliere & Tardella (1998) and Walker (2007), that, conditionally on $\rho_n$, we have $-\log(1 - V_i) \overset{iid}{\sim} \exp(\alpha)$ for $i > H_n$, which makes $\sum_{i=H_n+1}^{k} -\log(1 - V_i)$ the arrival times of a homogeneous Poisson process of rate $\alpha$. Hence, by its definition in (21), we have

$$
K_n^0(x) - H_n - 1 \mid \rho_n \sim \text{Poisson}\left(\alpha \log(1/x)\right) \tag{25}
$$

for any $x \in (0,1)$. Applying Lemma A.1, we have

$$
\mathbb{P}\left(K_n^0(x) - H_n \geq 1 + 3\alpha \log(1/x) + \log(2/\delta) \mid \rho_n\right) \leq \frac{\delta}{2} \tag{26}
$$

for any $x \in (0,1)$, and the bound is independent of $\rho_n$.

Now, for the second inequality in (24), we note that by Proposition 3.1 and Corollary 3.2

$$
\mathbb{P}\left(u_{\min} \leq x \mid \rho_n\right) \leq \mathbb{P}\left(u_{\min} \leq x \mid \rho^{\text{sing}}\right) \tag{27}
$$

where $\rho^{\text{sing}}$ is the singleton partition and $(c_1^{\text{sing}}, \ldots, c_n^{\text{sing}})$ a correspondent cluster-label vector, i.e., $c_i^{\text{sing}} \neq c_j^{\text{sing}}$ for any $i \neq j \in [n]$. Then, we use the following minimum argument and union bound

$$
\mathbb{P}\left(u_{\min} \leq x \mid \rho^{\text{sing}}\right) \leq \mathbb{P}\left(\bigcup_{i=1}^{n}\{u_i \leq x\} \,\Big|\, \rho^{\text{sing}}\right) \leq \sum_{i=1}^{n} \mathbb{P}\left(u_i \leq x \mid \rho^{\text{sing}}\right). \tag{28}
$$

Focusing on each summand on the r.h.s. of (28), let $w_i := \pi_{c_i^{\text{sing}}}$, i.e., the weight associated to the allocation of the $i$-th observation, so that the slice variable $u_i^{\text{sing}}$, conditionally on $w_i$, is uniform on $(0, w_i)$. In particular, for any $x \in (0,1)$ we have

$$
\mathbb{P}\left(u_i \leq x \mid w_i, \rho^{\text{sing}}\right) = \min\left(1, \frac{x}{w_i}\right).
$$

Moreover,

$$
\left(w_1, \ldots, w_n, 1 - \sum_{i=1}^{n} w_i\right) \,\Big|\, \rho^{\text{sing}} \sim \text{Dirichlet}(1, \ldots, 1, \alpha)
$$

and, in particular, $w_i \mid \rho^{\text{sing}} \sim \text{Beta}(1, \alpha + n - 1)$. Thus, for any $x \in (0,1)$

$$
\begin{aligned}
\mathbb{P}\left(u_i \leq x \mid \rho^{\text{sing}}\right) &= (\alpha + n - 1)\left[\int_0^x (1 - w)^{\alpha + n - 2}\, dw + x \int_x^1 \frac{(1 - w)^{\alpha + n - 2}}{w}\, dw\right] \\
&\leq (\alpha + n - 1)x\left[1 + \log(1/x)\right]
\end{aligned} \tag{29}
$$

where we use that $1 - w < 1$. Combining (27), (28) and (29) yields

$$
\mathbb{P}\left(u_{\min} \leq x \mid \rho_n\right) \leq n\left(\alpha + n - 1\right)x\left[1 + \log\left(1/x\right)\right]. \tag{30}
$$

We note that the bound is again independent of $\rho_n$.

Finally, we simply apply Lemma A.2 to choose $x = x(n, \delta) \in (0, 1)$ such that, for any $n \geq 2$, the lower-bound in the probability in (26) is upper-bounded by $C_\delta \log n$, and the r.h.s. of (30) is upper-bounded by $\delta/2$. In fact, going back to (26), from (14) we get

$$\mathbb{P}\left(K_n^0\left(x(n, \delta)\right) - H_n > C_\delta \log n \mid \rho_n\right) \leq \mathbb{P}\left(K_n^0(x(n, \delta)) - H_n > 1 + 3\alpha \log \frac{1}{x(n, \delta)} + \log \frac{2}{\delta} \,\Big|\, \rho_n\right) \leq \frac{\delta}{2}$$

which is the first bound in (24), while combining (30) and (13) gives the second one.

To sum up, we proved that, uniformly over any partition $\rho_n$ visited by the posterior algorithm at any considered iteration, for any $\delta \in (0, 1)$ there exists $C_\delta$ such that, for any $n \geq 2$, $\mathbb{P}\left(K_n - H_n > C_\delta \log n \mid \rho_n\right) \leq \delta$, where we give an explicit expression for the constant $C_\delta$ in (15). In particular, we have $K_n - H_n = O_\mathbb{P}(\log n)$ and in the worst-case scenario of $H_n = O(n)$, $K_n = O_\mathbb{P}(n)$. $\qquad\square$

### A.4. Proof of Corollary 3.4

Let $X_n := \frac{K_n - H_n}{\log n}$. In the bound obtained in Theorem 3.3 one can simply take $\delta = e^{-t}$ for $t > 0$ and obtain, for any $n \geq 2$,

$$\mathbb{P}\left(X_n > B_\alpha^{(1)} + B_\alpha^{(2)} t \,\Big|\, \rho_n\right) \leq e^{-t} \tag{31}$$

for any visited partition $\rho_n$, *i.e.*, an exponential bound on the tails of the normalized overhead, uniform in $n$ and in $\rho_n$. To obtain the uniform bound on the $p$-th moment of $X_n$, we leverage the tail moment identity: for any nonnegative random variable $Z$

$$\mathbb{E}\left[Z^p\right] = \int_0^\infty p s^{p-1} \mathbb{P}\left(Z > s\right) ds$$

which we apply for $Z = (X_n - B_\alpha^{(1)})_+$, obtaining

$$\begin{aligned}
\mathbb{E}\left[\left(X_n - B_\alpha^{(1)}\right)_+^p \,\Big|\, \rho_n\right] &= \int_0^\infty p s^{p-1} \mathbb{P}\left(X_n > B_\alpha^{(1)} + s \,\Big|\, \rho_n\right) ds \\
&\leq \int_0^\infty p s^{p-1} e^{-s/B_\alpha^{(2)}} ds = p\,\Gamma(p)\,(B_\alpha^{(2)})^p
\end{aligned} \tag{32}$$

where we applied the bound in (31) with $s = t B_\alpha^{(2)}$. Now we simply observe that

$$X_n = \left(X_n - B_\alpha^{(1)}\right)_+ + \min\left(X_n, B_\alpha^{(1)}\right),$$

and use the inequality $(a + b)^p \leq 2^{p-1}(a^p + b^p)$ valid for all $a, b \geq 0$ and $p \geq 1$. This yields

$$X_n^p \leq 2^{p-1}\left\{\left(X_n - B_\alpha^{(1)}\right)_+^p + \left(B_\alpha^{(1)}\right)^p\right\}$$

Hence, taking conditional expectation and using (32), we have

$$\mathbb{E}\left[X_n^p \mid \rho_n\right] \leq 2^{p-1}\left\{(B_\alpha^{(1)})^p + p\Gamma(p)\,(B_\alpha^{(2)})^p\right\} := C_{\alpha, p}$$

which, taking the supremum over $n \geq 2$, gives

$$\sup_{n \geq 2} \mathbb{E}\left[X_n^p \mid \rho_n\right] \leq C_{\alpha, p}$$

for any $\rho_n$.

### A.5. Proof of Corollary 3.5

Given the result in Theorem 3.3, taking expectation w.r.t. the partition $\rho_n$ we have

$$\mathbb{P}\left(K_n - H_n > \left[B_\alpha^{(1)} + B_\alpha^{(2)} \log(1/\delta_n)\right] \log n\right) \leq \delta_n \tag{33}$$

for any vanishing sequence $(\delta_n)_{n\geq 1} \subset (0, 1/2)$. Taking $D_\alpha := \frac{B_\alpha^{(1)}}{\log 2} + B_\alpha^{(2)}$ we have

$$\left[ B_\alpha^{(1)} + B_\alpha^{(2)} \log(1/\delta_n) \right] \log n \leq D_\alpha \log(1/\delta_n) \log n$$

since $\log 2 < \log(1/\delta_n)$. Hence (33) becomes

$$\mathbb{P}\left(K_n - H_n > D_\alpha \log(1/\delta_n) \log n\right) \leq \delta_n.$$

Finally, if $\sum_{n\geq 1} \delta_n < \infty$, then by Borel-Cantelli

$$\mathbb{P}\left(\limsup_{n\to\infty} \{K_n - H_n > D_\alpha \log(1/\delta_n) \log n\}\right) = 0.$$

### A.6. Proof of Proposition 3.6

First, we state and prove the following lemma.

**Lemma A.3.** *If* $(p_1, \ldots, p_n) \sim \mathrm{Dirichlet}(1, \ldots, 1)$*, then, for* $t \leq 1/n$*,* $\mathbb{P}(\min_{1\leq i\leq n} p_i > t) = (1 - nt)^{n-1}$*.*

*Proof of Lemma A.3.* We know that $(p_1, \ldots, p_n)$ is uniformly distributed on the simplex

$$\Delta_{n-1} := \left\{ (p_1, \ldots, p_n) : p_i \geq 0, \ \sum_{i=1}^{n} p_i = 1 \right\}.$$

For $t \leq 1/n$ the event $\{\min_{1\leq i\leq n} p_i > t\}$ corresponds to the set

$$A_t := \left\{ (p_1, \ldots, p_n) : p_i > t, \ \sum_{i=1}^{n} p_i = 1 \right\}.$$

Now, set

$$r_i = \frac{p_i - t}{1 - nt}, \qquad i = 1, \ldots, n$$

so that

$$A_t := \left\{ (p_1, \ldots, p_n) : p_i = t + (1 - nt)r_i \text{ and } r_i > 0, \ \sum_{i=1}^{n} r_i = 1 \right\}.$$

Thus $A_t$ is the image of $\Delta_{n-1}$ under the affine map

$$(r_1, \ldots, r_n) \mapsto \left(t + (1 - nt)r_1, \ldots, t + (1 - nt)r_n\right).$$

Since $\Delta_{n-1}$ has dimension $n-1$ and volume 1, the volume of $A_t$ is $(1 - nt)^{n-1}$. Therefore

$$\mathbb{P}\left(\min_{1\leq i\leq n} p_i > t \,\Big|\, \pi^\star, \rho_n^{\mathrm{sing}}\right) = (1 - nt)^{n-1}.$$

$\square$

We can now prove the tightness result.

*Proof of Proposition 3.6.* From the proof of Theorem 3.3, for any $b > 0$, we have $K_n^0(b/n) - H_n - 1 \mid \rho_n^{\mathrm{sing}} \sim$ Poisson $\left(\alpha \log \frac{n}{b}\right)$ and, in particular,

$$\mathbb{E}\left[K_n^0(b/n) - H_n - 1 \mid \rho_n^{\mathrm{sing}}\right] = \mathrm{Var}\left(K_n^0(b/n) - H_n - 1 \mid \rho_n^{\mathrm{sing}}\right) = \alpha \log \frac{n}{b}.$$

Therefore, for any $c_\alpha \in (0, \alpha)$, for $n$ large enough, $c_\alpha \log n < \alpha \log \frac{n}{b}$ and by Chebyshev's inequality,

$$
\begin{aligned}
\mathbb{P}\left(K_n^0(b/n) - H_n < c_\alpha \log n \,\middle|\, \rho_n^{\text{sing}}\right) &= \mathbb{P}\left(K_n^0(b/n) - H_n - 1 < c_\alpha \log n - 1 \,\middle|\, \rho_n^{\text{sing}}\right) \\
&\leq \mathbb{P}\left(\left|K_n^0(b/n) - H_n - 1 - \alpha \log \frac{n}{b}\right| \geq \alpha \log \frac{n}{b} - c_\alpha \log n - 1 \,\middle|\, \rho_n^{\text{sing}}\right) \\
&\leq \frac{\alpha \log(n/b)}{(\alpha \log(n/b) - c_\alpha \log n - 1)^2} = \frac{\alpha \log n - \alpha \log b}{((\alpha - c_\alpha) \log n - \alpha \log b - 1)^2} \to 0.
\end{aligned}
$$

That is, $\mathbb{P}\left(K_n^0(b/n) - H_n \geq c_\alpha \log n \,\middle|\, \rho_n^{\text{sing}}\right) \to 1$ and, in particular, for all sufficiently large $n$,

$$
\mathbb{P}\left(K_n^0(b/n) - H_n \geq c_\alpha \log n \,\middle|\, \rho_n^{\text{sing}}\right) \geq \frac{1}{2}.
$$

Now consider the event $E_n = \left\{\pi^\star \geq \frac{a}{n}\right\} \cap \left\{u_{\min} \leq \frac{ab}{n^2}\right\}$ for some $a > 0$. Note that, from the proof of Theorem 3.3, we have $K_n = K_n^0(u_{\min}/\pi^\star)$, and thus $K_n \geq K_n^0(b/n)$ on $E_n$. Moreover, conditionally on $\rho_n^{\text{sing}}$, $K_n^0(b/n)$ depends only on $(V_i)_{i > H_n}$, while $E_n$ depends only on $(\pi_1, \ldots, \pi_H, \pi^\star)$ and $(u_i)$, which are independent by construction; hence $E_n$ and $K_n^0(b/n)$ are conditionally independent. Thus, for $n$ large enough,

$$
\mathbb{P}(K_n - H_n \geq c_\alpha \log n \mid \rho_n^{\text{sing}}) \geq \mathbb{P}(E_n \mid \rho_n^{\text{sing}})\mathbb{P}(K_n^0(b/n) - H_n \geq c_\alpha \log n \mid \rho_n^{\text{sing}}) \geq \frac{1}{2}\mathbb{P}(E_n \mid \rho_n^{\text{sing}}).
$$

It remains to show that $\mathbb{P}(E_n \mid \rho_n^{\text{sing}})$ is bounded away from 0. Since $\pi^\star \sim \text{Beta}(\alpha, n)$, we have $n\pi^\star \xrightarrow{d} \text{Gamma}(\alpha, 1)$, so, for any fixed $a > 0$,

$$
\mathbb{P}\left(\pi^\star \geq \frac{a}{n} \,\middle|\, \rho_n^{\text{sing}}\right) \to 1 - F(a) > 0
$$

where $F(a)$ denotes the cdf of a $\text{Gamma}(\alpha, 1)$. Moreover, let $p_i = \pi_{c_i}/(1 - \pi^\star)$, then $(p_1, \ldots, p_n) \mid \pi^\star \sim \text{Dirichlet}(1, \ldots, 1)$ and by Lemma A.3, for $t \leq 1/n$,

$$
\mathbb{P}\left(\min_{1 \leq i \leq n} p_i > t \,\middle|\, \pi^\star, \rho_n^{\text{sing}}\right) = (1 - nt)^{n-1}.
$$

Now note that $\min_{1 \leq i \leq n} \pi_{c_i} = (1 - \pi^\star) \min_{1 \leq i \leq n} p_i$. Therefore

$$
\mathbb{P}\left(\min_{1 \leq i \leq n} \pi_{c_i} \leq \frac{ab}{n^2} \,\middle|\, \pi^\star, \rho_n^{\text{sing}}\right) = \mathbb{P}\left(\min_{1 \leq i \leq n} p_i \leq \frac{ab}{(1 - \pi^\star)n^2} \,\middle|\, \pi^\star, \rho_n^{\text{sing}}\right) \geq \mathbb{P}\left(\min_{1 \leq i \leq n} p_i \leq \frac{ab}{n^2} \,\middle|\, \pi^\star, \rho_n^{\text{sing}}\right).
$$

Taking $t = ab/n^2$, we obtain

$$
\mathbb{P}\left(\min_{1 \leq i \leq n} p_i \leq \frac{ab}{n^2} \,\middle|\, \pi^\star, \rho_n^{\text{sing}}\right) = 1 - \left(1 - \frac{ab}{n}\right)^{n-1} \to 1 - e^{-ab} > 0.
$$

Now, let $i^\star = \operatorname*{argmin}_{1 \leq i \leq n} \pi_{c_i}$. Since $u_{i^\star} \mid \pi_{c_{i^\star}} \sim \text{Unif}(0, \min_{1 \leq i \leq n} \pi_{c_i})$, we have

$$
\mathbb{P}\left(u_{\min} \leq \frac{ab}{n^2} \,\middle|\, \min_{1 \leq i \leq n} \pi_{c_i} \leq \frac{ab}{n^2}\right) = 1.
$$

Therefore

$$
\mathbb{P}\left(u_{\min} \leq \frac{ab}{n^2} \,\middle|\, \pi^\star, \rho_n^{\text{sing}}\right) \geq \mathbb{P}\left(\min_{1 \leq i \leq n} \pi_{c_i} \leq \frac{ab}{n^2} \,\middle|\, \pi^\star, \rho_n^{\text{sing}}\right) > 0 \qquad \mathbb{P}(\cdot \mid \rho_n^{\text{sing}})\text{-a.s.}
$$

Combining the previous displays, there exists $\eta_{a,b} > 0$ such that for all sufficiently large $n$,

$$
\mathbb{P}\left(u_{\min} \leq \frac{ab}{n^2} \,\middle|\, \pi^\star, \rho_n^{\text{sing}}\right) \geq \eta_{a,b} \qquad \mathbb{P}(\cdot \mid \rho_n^{\text{sing}})\text{-a.s.}
$$

To conclude, note that

$$
\mathbb{P}\left(E_n \mid \rho_n^{\text{sing}}\right) = \mathbb{E}\left[\mathbf{1}_{\{\pi^\star \geq a/n\}} \mathbb{P}\left(u_{\min} \leq \frac{ab}{n^2} \,\middle|\, \pi^\star, \rho_n^{\text{sing}}\right) \,\middle|\, \rho_n^{\text{sing}}\right].
$$

and thus, for $n$ large enough, $\mathbb{P}\left(E_n \mid \rho_n^{\text{sing}}\right) \geq \eta_{ab}\left(1 - F(a)\right)$, where the right-hand side is bounded away from 0.

The second claim follows immediately. $\qquad \square$

# B. Additional Details on Numerical Experiments

## B.1. Three Equally-balanced Clusters

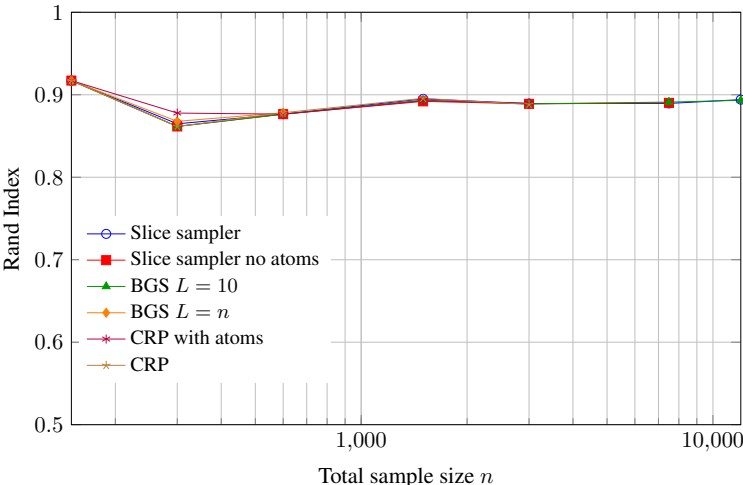

*Figure 5.* **Three equally-balanced clusters**: Rand index between the partition's point estimate and the true clustering configuration. The point estimate is obtained by minimizing the Binder loss function.

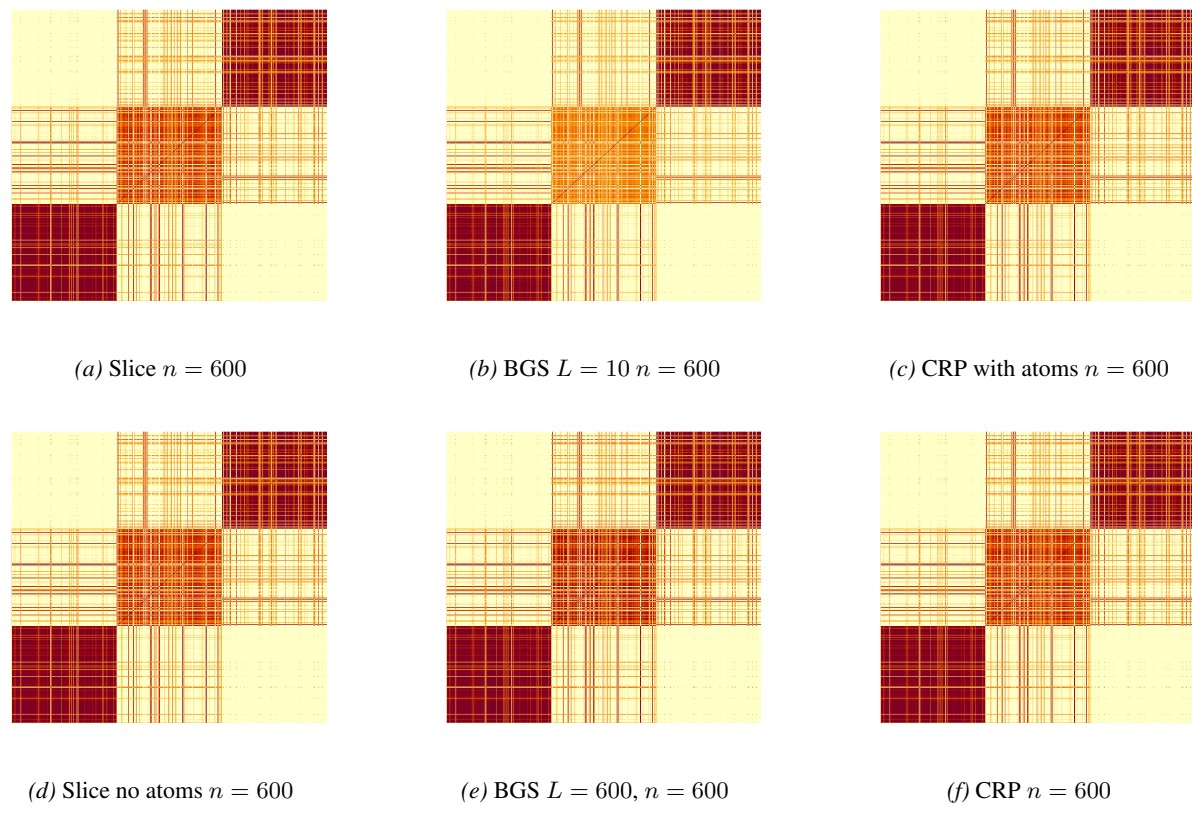

*(a)* Slice $n = 600$      *(b)* BGS $L = 10$ $n = 600$      *(c)* CRP with atoms $n = 600$

*(d)* Slice no atoms $n = 600$      *(e)* BGS $L = 600, n = 600$      *(f)* CRP $n = 600$

*Figure 6.* **Three equally-balanced clusters**: Posterior co-clustering matrices, computed discarding 5,000 iterations of burn-in.

## B.2. Perturbed Zipf

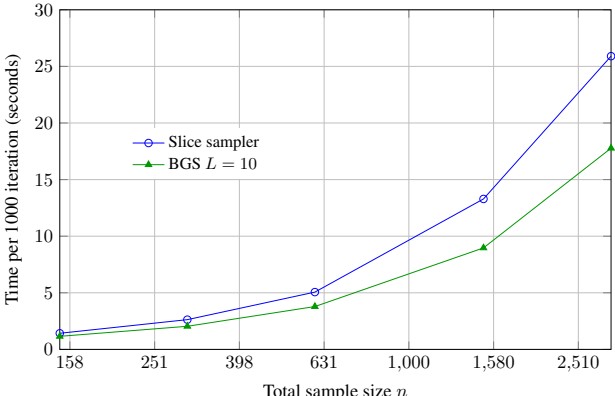

*Figure 7.* **Perturbed Zipf–generated data**: Time per 1000 iterations in seconds computed as average over 10,000 iterations (including burn-in) as a function of input size (x-axis on log scale). Codes are in R and run on a laptop with 13th Gen Intel(R) Core(TM) i7-1370P.

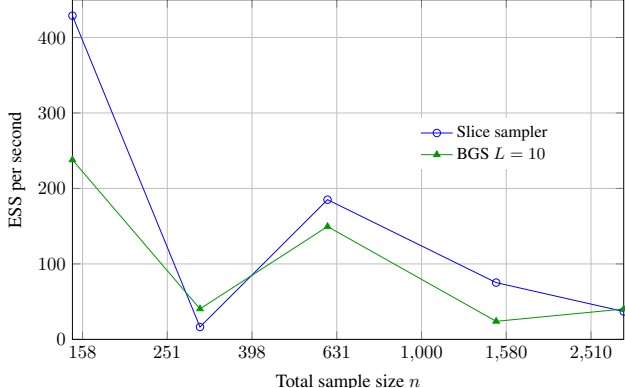

*Figure 8.* **Perturbed Zipf–generated data**: ESS (w.r.t. the log likelihood) per second computed as average over the last 5,000 iterations (excluding a burn-in of 5,000) as a function of input size (x-axis on log scale). Codes are in R and run on a laptop with 13th Gen Intel(R) Core(TM) i7-1370P.

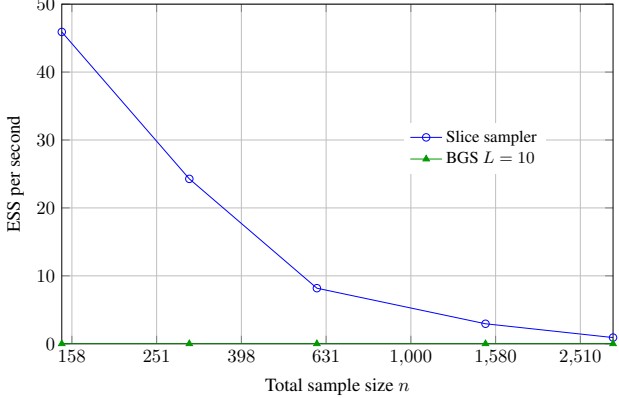

*Figure 9.* **Perturbed Zipf–generated data**: ESS (w.r.t. the number of clusters) per second computed as average over the last 5,000 iterations (excluding a burn-in of 5,000) as a function of input size (x-axis on log scale). Codes are in R and run on a laptop with 13th Gen Intel(R) Core(TM) i7-1370P.

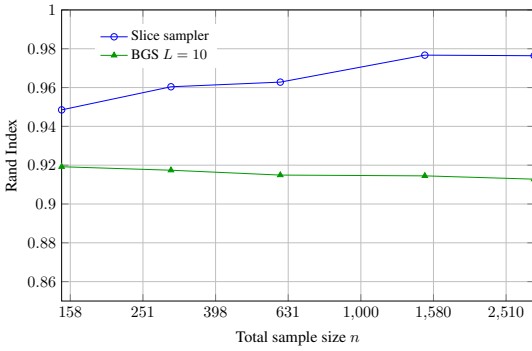

*Figure 10.* **Perturbed Zipf–generated data**: Rand index between the partition's point estimate and the true clustering configuration. The point estimate is obtained by minimizing the Binder loss function after discarding the first 5,000 iterations as burn-in.

| | | |
|---|---|---|
| *(a)* Truth $n = 150$ | *(b)* Truth $n = 300$ | *(c)* Truth $n = 600$ |
| *(d)* Slice $n = 150$ | *(e)* Slice $n = 300$ | *(f)* Slice $n = 600$ |
| *(g)* BGS $n = 150$ | *(h)* BGS $n = 300$ | *(i)* BGS $n = 600$ |

*Figure 11.* **Perturbed Zipf–generated data**: Posterior co-clustering matrices, computed discarding 5,000 iterations of burn-in.

## B.3. Results with Fixed Concentration Parameter

*Table 2.* **Three equally-balanced clusters**: computational and clustering performance results, $\alpha = 1$.

| Metric | | $n = 150$ | $n = 300$ | $n = 600$ | $n = 1500$ | $n = 3000$ | $n = 7500$ | $n = 12000$ |
|---|---|---|---|---|---|---|---|---|
| Time 1000 iter (sec.) | Slice | 1.382 | 2.332 | 4.372 | 10.295 | 21.570 | 65.948 | 191.935 |
| | Slice no atoms | 1.616 | 2.914 | 5.303 | 12.707 | 27.059 | 73.525 | NA |
| | BGS $L = 10$ | 1.275 | 2.213 | 4.158 | 9.790 | 20.188 | 59.433 | 106.416 |
| | BGS $L = n$ | 4.021 | 13.704 | 59.799 | NA | NA | NA | NA |
| | CRP w atoms | 2.198 | 4.415 | 10.141 | 22.485 | NA | NA | NA |
| | CRP | 3.279 | 6.942 | 14.435 | 36.775 | 74.377 | NA | NA |
| ESS per second (lik) | Slice | 474.371 | 428.634 | 147.212 | 54.093 | 18.107 | 3.248 | 9.810 |
| | Slice no atoms | 302.438 | 82.285 | 83.637 | 16.000 | 4.387 | 7.248 | NA |
| | BGS $L = 10$ | 467.188 | 118.002 | 165.963 | 16.529 | 11.848 | 7.010 | 2.766 |
| | BGS $L = n$ | 145.714 | 24.041 | 12.170 | NA | NA | NA | NA |
| | CRP w atoms | 274.589 | 73.424 | 73.040 | 8.608 | NA | NA | NA |
| | CRP | 222.056 | 63.014 | 51.481 | 6.290 | 5.883 | NA | NA |
| RI | Slice | 0.909 | 0.858 | 0.878 | 0.894 | 0.888 | 0.890 | 0.893 |
| | Slice no atoms | 0.909 | 0.864 | 0.876 | 0.894 | 0.887 | 0.891 | NA |
| | BGS $L = 10$ | 0.909 | 0.861 | 0.878 | 0.891 | 0.887 | 0.890 | 0.893 |
| | BGS $L = n$ | 0.917 | 0.861 | 0.876 | NA | NA | NA | NA |
| | CRP w atoms | 0.909 | 0.864 | 0.876 | 0.894 | NA | NA | NA |
| | CRP | 0.917 | 0.858 | 0.876 | 0.893 | 0.887 | NA | NA |
| ARI | Slice | 0.796 | 0.682 | 0.725 | 0.761 | 0.748 | 0.753 | 0.759 |
| | Slice no atoms | 0.796 | 0.696 | 0.721 | 0.763 | 0.747 | 0.755 | NA |
| | BGS $L = 10$ | 0.796 | 0.689 | 0.725 | 0.754 | 0.747 | 0.753 | 0.761 |
| | BGS $L = n$ | 0.812 | 0.690 | 0.721 | NA | NA | NA | NA |
| | CRP w atoms | 0.796 | 0.696 | 0.721 | 0.762 | NA | NA | NA |
| | CRP | 0.812 | 0.682 | 0.721 | 0.760 | 0.745 | NA | NA |

*Table 3.* **Perturbed Zipf–generated data**: computational and clustering performance results, $\alpha = 1$.

| Metric | | $n = 150$ | $n = 300$ | $n = 600$ | $n = 1500$ | $n = 3000$ |
|---|---|---|---|---|---|---|
| Time 1000 iter (sec.) | Slice | 1.451 | 2.604 | 5.050 | 13.391 | 26.077 |
| | BGS | 1.163 | 2.047 | 3.788 | 9.119 | 17.915 |
| ESS per second (lik) | Slice | 431.487 | 13.230 | 98.492 | 50.619 | 38.348 |
| | BGS $L = 10$ | 236.193 | 41.905 | 142.874 | 21.124 | 39.619 |
| ESS per second ($H$) | Slice | 33.189 | 15.757 | 9.862 | 2.307 | 0.996 |
| | BGS $L = 10$ | 0.000 | 0.000 | 0.000 | 0.000 | 0.000 |
| RI | Slice | 0.949 | 0.960 | 0.966 | 0.976 | 0.976 |
| | BGS | 0.920 | 0.917 | 0.915 | 0.914 | 0.913 |
| ARI | Slice | 0.450 | 0.540 | 0.587 | 0.630 | 0.646 |
| | BGS | 0.350 | 0.353 | 0.374 | 0.347 | 0.352 |

