# OpenReview forum: "Complexity Bounds for Dirichlet Process Slice Samplers"
_ICML.cc/2026/Conference — ICML 2026 regular_

### Official Review · Reviewer_A8v2 · 2026-03-11

**Soundness:** 4
**Presentation:** 4
**Significance:** 1
**Originality:** 3
**Overall Recommendation:** 4
**Confidence:** 5

**Summary:**

This is a mostly theoretical Bayesian nonparametric paper. The main result is a high probability bound on the number of instantiated clusters in a slice sampler for Dirichlet process mixture models. The importance of the result is that it shows that the slice sampler will, with high probability, only need log n more clusters than the number of occupied clusters. Some experimental results show the effectiveness and efficiency of the slice sampler relative to other samplers for DP mixtures (though the algorithm itself was an existing one and isn't a novel contribution). The paper is well- and carefully-written, there are no technical issues I find with it.

**Compliance With Llm Reviewing Policy:**

Affirmed.

**Key Questions For Authors:**

I have no questions for the authors, the paper is very well-written and the ideas clearly set out.

**Limitations:**

I don't see any limitations or negative societal impacts.

**Strengths And Weaknesses:**

Soundness: the paper is technically sound, and claims are sensible and well supported. Proofs are in appendix and look reasonable.

Presentation: well-written paper.

Originality: the new insight is that it gives the community theory-backed confidence that the number of instantiated clusters in the slice sampler wouldn't "blow up". This is a nice to have, though unsurprising given the practical and methodological understanding.

Significance: the weakest point of the paper is the significance. I think it is a useful and important result for Bayesian nonparametric practitioners, but since 15 years ago, the machine learning community has moved on from those ideas to work on other ideas. While Bayesian nonparametrics is still an important constellation of ideas in computational statistics and Bayesian statistics, their significance in machine learning is little now, and there is little applied impact happening in machine learning.

---

> ### Author Rebuttal · Authors · 2026-03-30
>
> We thank the reviewer for the careful reading and for the overall positive feedback.
>
> We appreciate the concern regarding significance in the context of current machine learning trends, and we would like to clarify two aspects that strengthen the broader impact of the contribution.
>
> First, while Bayesian nonparametric models may not be predominant in contemporary machine learning research, Dirichlet process–based constructions remain a competitive and central tool in probabilistic modeling, particularly in clustering, network analysis, and flexible latent variable models. In this context, our work provides a theoretical foundation for a widely used inference mechanism in these models.
>
> Second, beyond the specific DP setting, our contribution can be viewed as a first step toward deriving probabilistic guarantees for a broader class of probabilistic models with more general adaptive latent structure, while some of these extensions may not be trivial. In many modern approaches, the number of active latent components is data-dependent and must be controlled as part of the inference procedure, as in adaptive mixtures or gating-based models. The techniques developed in this work provide a principled way to analyze this aspect and offer a starting point for extending such guarantees beyond Dirichlet process models.

---

> > ### Author Rebuttal · Reviewer_A8v2 · 2026-03-31
> >
> > I am happy to recommend accepting the paper. But have to say am still not fully convinced of the significance on the community as a whole.

---

> > > ### Author Response · Authors · 2026-04-07
> > >
> > > Thank you for your thorough review and constructive criticism. We greatly appreciate the feedback and the positive recommendation for acceptance.

---

### Official Review · Reviewer_sebZ · 2026-03-11

**Soundness:** 4
**Presentation:** 3
**Significance:** 4
**Originality:** 4
**Overall Recommendation:** 6
**Confidence:** 3

**Summary:**

The paper studies posterior sampling in Dirichlet process generative models using slice sampling based on the stick-breaking representation of the Dirichlet process. In Walker's construction (2017), the slice variables induce a random truncation level $K$, so that only the first $K$ stick-breaking components need to be generated at a given iteration. The computational cost of Algorithm 2 is therefore governed by the truncation level $K$. To the best of the reviewer's knowledge, this appears to be the first paper to derive an explicit high-probability complexity bound for the dynamic truncation level of a DP slice sampler.

The central technical difficulty is that $K$ depends on the minimum slice variable $u_{\min}$: larger values of $u_{\min}$ lead to smaller truncation levels and hence cheaper iterations (Equation (2) in the paper). The main challenge is therefore to make sure $u_{\min}$ is not too small.

**Compliance With Llm Reviewing Policy:**

Affirmed.

**Key Questions For Authors:**

The reviewer has no additional critical questions for the authors at this time. The paper is generally clear and the main technical arguments appear sound. The comments above are mainly suggestions for improving the presentation (e.g., providing more intuition for Proposition 3.1 and restructuring parts of the proof of Theorem 3.3) rather than issues that would affect the evaluation of the work.

**Limitations:**

Yes

**Strengths And Weaknesses:**

Soundness: The paper is mathematically solid overall. The proof arguments seem to be carefully done and sufficiently rigorous.

Presentation: The paper is generally well written, but readability could be improved in a few places. First, Proposition 3.1 about cluster merging is the key technical input of the paper, so the main text would benefit from a short discussion of the proof strategy, especially the roles of equations (4) and (8). Second, the proof of Theorem 3.3 is rather long. The argument from (14) to (19) could be isolated as a short lemma, and the argument from (20) to (23) could be stated as a separate lemma also. This would make the proof of Theorem 3.3 easier to follow.

Significance: The result is significant as it is the first paper to provide theoretical complexity results about slice samplers.

Originality: The originality is excellent. The key idea is Proposition 3.1, which shows that merging two clusters makes very small values of $u_{\min}$ less likely. By iterating this argument from Proposition 3.1, Corollary 3.2 identifies the singleton partition (all clusters are singletons) as the worst-case configuration. This is the key idea of the paper: it reduces the analysis over all possible partitions to a single worst case. That reduction makes the derivation of the main complexity bound tractable.

---

> ### Author Rebuttal · Authors · 2026-03-30
>
> We thank the reviewer for the helpful suggestions that we decided to follow to improve readability and highlight the intuition behind the most technical parts of the work. In particular, in the final version of the paper, we externalize the technical parts of the proof of the main Theorem in the following two lemmas.
>
> Lemma A.1.
> Let $Z\sim\mathrm{Possion}(\alpha\log(1/x))$ for any $\alpha>0$ and any $x\in(0,1)$. Then \begin{equation}\mathbb P\left(Z\geq 3\alpha\log(1/x)+\log(2/\delta)\right)\leq\frac{\delta}{2}\end{equation}
>
> Lemma A.2. For any $n\geq 2$, $\alpha>0$, and $\delta\in(0,1)$, if
>     \begin{equation}
>     x(n,\delta):=\frac{\delta}{4n(\alpha+n-1)\log(2n(\alpha+n-1)e/\delta)}
> \end{equation}
> then we have
> \begin{equation}n(\alpha+n-1)x(n,\delta)[1+\log(1/x(n,\delta))]\le\frac{\delta}{2}\end{equation}
> and
> \begin{equation}
> 1+3\alpha\log(1/x(n,\delta))+\log(2/\delta)\le C_\delta\log n\end{equation}
> where
> \begin{equation}
>    C_\delta
> =
> B_\alpha^{(1)}
> +
> B_\alpha^{(2)}\log\frac{1}{\delta}\quad
> \text{with}\quad
> B_\alpha^{(2)}=\frac{6\alpha+1}{\log 2},\quad
> B^{(1)}_\alpha
> =
> 12\alpha
> +
> \frac{
> 1
> +3\alpha\log\big(8e(1+\alpha)^2\big)
> +\log 2
> }{\log 2}.
> \end{equation}
> Moreover, we add the following after Proposition 3.1 in the main text:
> ``Intuitively, Proposition 3.1 reflects the fact that splitting a cluster into two divides the associated weight, yielding smaller weights and hence narrower supports for the corresponding slice variables. This reduces the probability that slice variables exceed a given threshold."

---

> > ### Author Rebuttal · Reviewer_sebZ · 2026-04-02
> >
> > I will maintain my rating and am happy to recommend acceptance.

---

> > > ### Author Response · Authors · 2026-04-07
> > >
> > > Thank you for your review and for helping us improve the presentation of the results. We greatly appreciate the positive recommendation for acceptance.

---

### Official Review · Reviewer_LMkb · 2026-03-12

**Soundness:** 3
**Presentation:** 3
**Significance:** 3
**Originality:** 3
**Overall Recommendation:** 5
**Confidence:** 3

**Summary:**

The paper provides a significant theoretical contribution by establishing the first a posteriori high-probability complexity guarantees for slice samplers in Dirichlet Process (DP) mixture models. While slice samplers are widely used for exact posterior simulation without fixed truncation, their computational scalability remained theoretically uncertain due to the random nature of the per-iteration cost. This work addresses this gap by proving that the number of sampled components beyond those occupied by data is at most logarithmic in the sample size ($O_P(\log n)$), regardless of the data partition. This result offers a formal justification for the empirical scalability of slice-based approaches in high-dimensional or large-scale settings.

**Compliance With Llm Reviewing Policy:**

Affirmed.

**Final Justification:**

The rebuttal has strengthened the validity of the research. Consequently, I will raise the Soundness score from 2 to 3 and change the Overall Recommendation from 4 to 5.

**Key Questions For Authors:**

[1] Re-evaluation of clustering performance (ARI):
In Figure 4, the synthetic Zipf model creates a regime with many small clusters. In such settings, the standard Rand Index (RI) tends to be inflated by the high frequency of "true negatives" (pairs correctly assigned to different clusters by chance). To ensure the robustness of your claims regarding the sampler's accuracy, please provide the Adjusted Rand Index (ARI) or Variation of Information (VI) for these results. Demonstrating that the slice sampler maintains high performance under these chance-corrected metrics would significantly strengthen the paper’s empirical validity.

Optionanal:
[2] Impact of a lower bound $\delta$ on the slice variable:
From a practical implementation standpoint, one might consider restricting the slice variable to $u \sim \text{Uniform}(\delta, f(x))$ to ensure numerical stability and prevent rare computational spikes. Based on your $O_P(\log n)$ framework, could you provide any theoretical insights or intuition on how such a $\delta$-truncation might affect the balance between computational gain and the resulting posterior bias?

**Limitations:**

yes

**Strengths And Weaknesses:**

Soundness:

The mathematical derivation appears solid, particularly the establishment of exponential tails for the complexity bounds, which provides strong insurance for the algorithmic stability in practice.

There is a concern regarding the evaluation metrics used in the numerical experiments. In Figure 4 (Zipf model), the authors employ the Rand Index (RI). Given that this scenario involves a large number of small clusters, RI may be prone to a high baseline score due to chance agreement on "different-cluster" assignments, potentially masking the true inferential performance.

Presentation:

The paper is well-structured and clearly written. It successfully guides the reader from the motivation and the limitations of fixed truncation to the sophisticated probabilistic bounds of the slice sampler.

Significance:

This work reconciles the trade-off between "exact targeting" and "computational scalability." By proving that the exact slice sampler can be as efficient as biased truncation methods (on an order-of-magnitude basis), the paper provides a strong theoretical mandate for using exact methods in large-scale Bayesian nonparametrics.

Originality:
The approach of analyzing slice sampler efficiency through the lens of posterior uniformity is novel. By identifying the "singleton partition" as the worst-case scenario for the slice variable $u_{\min}$ and leveraging the properties of Dirichlet-Multinomial distributions, the authors provide a rigorous foundation that complements existing literature on prior-based analysis.

---

> ### Author Rebuttal · Authors · 2026-03-30
>
> $[1]$ Thank you for this comment.
>
> We initially reported the Rand Index (RI) because it is directly interpretable on the [0,1] scale. However, you are right that in regimes characterized by many small clusters, such as the Zipf setting in Figure 4, the RI can be inflated by the large number of true negatives and may therefore be misleading. Including a chance-corrected metric is therefore important to strengthen the empirical evaluation.
>
> Following your suggestion, we now also compute and report the Adjusted Rand Index (ARI), and include the corresponding box plots for all partitions visited by the chains. The median ARI values, reported alongside the median RI for ease of comparison, are shown below:
>
> RI (old - Figure 4)
> | Algorithm | 150   | 300   | 600   | 1500  | 3000  |
> |-----------|-------|-------|-------|-------|-------|
> | Slice     | 0.870 | 0.886 | 0.926 | 0.920 | 0.900 |
> | BGS       | 0.812 | 0.586 | 0.618 | 0.553 | 0.600 |
>
> ARI (new)
> | Algorithm | 150   | 300   | 600   | 1500  | 3000  |
> |-----------|-------|-------|-------|-------|-------|
> | Slice     | 0.740 | 0.773 | 0.849 | 0.835 | 0.819 |
> | BGS       | 0.635 | 0.262 | 0.310 | 0.201 | 0.224 |
>
> These results confirm that the slice sampler maintains strong performance under a chance-corrected metric. Importantly, the qualitative conclusions remain unchanged: the slice sampler consistently achieves high clustering accuracy, while the performance of BGS deteriorates rapidly as the sample size increases in the Zipf setting.
>
> $[2]$ We thank the reviewer for suggesting this interesting direction. We provide here some comments, highlighting how our work can help in quantifying the suitability and gains of this alternative strategy. We replace the notation $\delta$ with $\epsilon$ to avoid confusion with our high-probability parameter. To obtain an operative implementation of the suggested strategy, we consider the variant of Algorithm 2 where slicing variables are sampled as $u_i \sim \mathrm{Uniform}(\epsilon, \pi_{c_i})$, if $\epsilon<\pi_{c_i}$, and $u_i\overset{a.s.}{=}\epsilon$ otherwise.
>
> In terms of computational complexity, our main theorem can be used to derive the following results on the proposed variant. For a fixed $\epsilon$ (constant in $n$), the overhead remains random, and following our proof techniques, we obtain the bound
> $$\text{Pr}(K_n - H_n \geq 1 + 3\alpha \log(1/\epsilon) + \log(1/\delta)) < \delta,$$
> showing that the overhead becomes $O_\mathbb P(1)$ rather than $O_\mathbb P(\log n)$, obviously at the cost of introducing a truncation bias.
>
> Indeed, it is reasonable to expect that a constant-in-$n$ $\epsilon$ induces increasing bias as the sample size grows, especially in regimes where the number of visited clusters increases with $n$ (as in our second numerical study).
>
> To investigate this, we implemented the truncated scheme for $\epsilon \in (0.01, 10^{-3}, 10^{-4}, 10^{-10}, 10^{-100})$ and compared it with the exact slice sampler and BGS for $n=600$, for both simulation studies considered in our work. The results reported below are consistent with this intuition, where in the second simulation study, significant gains in computational time are achieved only at the cost of important inferential performance degradation.
>
> | Algorithm   | time (sim 1) | time (sim 2) | ARI (sim 1) | ARI (sim 2) |
> |-------------|--------------|--------------|-------------|-------------|
> | Slice       | 4.22         | 2.88         | 0.70        | 0.85        |
> | BGS 10      | 3.97         | 3.93         | 0.70        | 0.31        |
> | Appr 0.01   | 3.98         | 0.85         | 0.70        | 0.05        |
> | Appr 1e-03  | 4.00         | 0.93         | 0.70        | 0.60        |
> | Appr 1e-04  | 4.09         | 2.76         | 0.70        | 0.85        |
> | Appr 1e-10  | 4.10         | 2.79         | 0.70        | 0.85        |
> | Appr 1e-100 | 4.10         | 2.84         | 0.70        | 0.85        |
>
> It's clear from its construction that, for $\epsilon\to0$, the variant tends to our original Algorithm 2. Therefore, we believe a promising direction for future work is to let $\epsilon$ vary with $n$, and to derive principled rates balancing the trade-off between computational overhead and truncation bias. Our current contribution provides ready-to-use tools to quantify the former, while the latter remains an open and interesting problem.

---

> > ### Author Rebuttal · Reviewer_LMkb · 2026-04-03
> >
> > The additional verification using ARI has further highlighted the superiority of the proposed method over BGS, thereby strengthening the validity of the research.
> > The optional questions were also addressed with effective discussion and verification.
> > I will change my rating from 4 to 5.

---

> > > ### Author Response · Authors · 2026-04-07
> > >
> > > Thank you for your careful and thorough review. We also found your second question particularly interesting and appreciate you highlighting this direction. We greatly appreciate the positive recommendation for acceptance.

---

### Official Review · Reviewer_U5pP · 2026-03-12

**Soundness:** 3
**Presentation:** 3
**Significance:** 3
**Originality:** 3
**Overall Recommendation:** 5
**Confidence:** 2

**Summary:**

This paper provides the first high-probability complexity guarantees for slice sampling in Dirichlet process–based models, showing that the dynamic truncation level​ exceeds the number of occupied clusters by at most $O_P(\log n)$, uniformly over all partitions visited by the posterior chain. The proof combines a worst-case reduction to the singleton partition with Poisson tail bounds and a union bound on the minimum slice variable. Numerical experiments on DP mixture models confirm the favorable scaling and demonstrate that slice samplers maintain accurate clustering in growing-cluster regimes where fixed-truncation blocked Gibbs samplers fail.

**Compliance With Llm Reviewing Policy:**

Affirmed.

**Final Justification:**

Since the authors have addressed my concerns, I would be happy to recommend acceptance.

**Key Questions For Authors:**

- Is the rate $log n$ tight? could the true overhead be $O(1)$ or $O(\log\log n)$ in typical posterior regimes?

- The theory is stated for fixed $\alpha$, whereas the experiments place a prior on $\alpha$. Could the authors clarify this?

**Limitations:**

yes

**Strengths And Weaknesses:**

Strength:

- This paper provides a formal analysis of the computational complexity of slice sampling for a broad class of DP–based models and fills a open problem.

- Uniformity over partitions is nontrivial and practically meaningful and Proposition 3.1 reduces the problem to the singleton worst case, and may have independent value.

- this paper is well-written and the proof is clear.

Weakness:

- There is no mixing time analysis and only per-iteration cost was studied.

---

> ### Author Rebuttal · Authors · 2026-03-30
>
> 1. Tightness.
>
> Thank you for this question. We believe that, for clarity, it is useful to address the two parts of this question separately. As a uniform upper-bound of the overhead, $\log n$ is tight, meaning that there is no slower rate that would do the trick uniformly over all the partitions. In fact, building on the arguments used in the proof of our main theorem, we now also show the following.
>
> Proposition new. Let $(K_n, H_n)$ be defined as in Theorem 3.3.
> Then there exist constants $c_\alpha,\eta_\alpha>0$ such that for all sufficiently large $n$,
> $$\mathbb P \left(K_n - H_n \ge c_\alpha \log n | \rho_n^{\mathrm{sing}}\right)\ge \eta_\alpha.$$
> Consequently, for any deterministic sequence $r_n = o(\log n)$ and any $M>0$,
> $$\liminf_{n\to\infty} \sup_{\rho_n}
> \mathbb P\left(K_n - H_n > M r_n | \rho_n\right) > 0.$$
>
> In other words, under the singleton partition, for any $r_n = o(\log n)$, $K_n - H_n$ is not $O_\mathbb P(r_n)$.
> As a consequence, any rate slower than $\log n$ would not uniformly upper-bound the overhead.
>
> To address the second part of the question, the discussion above does not imply that, when conditioning on a specific partition, the rate of the overhead cannot be slower. We conjecture that the tightness of the $\log n$ rate for a given partition is closely related to the presence of singleton clusters. For intermediate regimes, e.g., $H_n \sim \log n$, the rate may or may not be tight depending on the smallest cluster sizes. Nonetheless, even posteriors that are highly concentrated on “simple” partitions may still assign non-negligible mass to more “complex” configurations, or the Markov chain may need to visit them before convergence.
>
> In conclusion, since the posterior partition structure is difficult to characterize before running the sampler and since the chain may need to explore also partitions with small posterior mass, we believe that uniform guarantees over all partitions that could be visited are the most meaningful in practice. For such uniform guarantees, our rate is tight. While improved rates may be possible when restricting attention to specific partitions, such refinements would likely require strong assumptions that are difficult to justify in practice.
> We now include comments regarding tightness and the new result in the paper.
>
> 2. Randomness of $\alpha$.
>
> Thank you for this question. The theoretical results are indeed stated conditionally on a fixed value of $\alpha$. Importantly, our analysis yields constants that are explicit in $\alpha$, which allows the results to be extended to settings where $\alpha$ is random or data-driven. Such extensions would require specifying a particular prior or tuning scheme for $\alpha$, and the resulting bounds would depend on that choice. We view this as a natural direction for future work.
>
> Regarding the experiments, we agree that considering a fixed value of $\alpha$ is relevant. While we initially used a Gamma prior to reflect common practice in mixture modeling, we have rerun the experiments with $\alpha=1$ fixed and will also include these results. The empirical behavior is consistent with the original findings: in particular, scalability (wall-clock time), mixing properties (ESS per sec), and inferential performance (RI/ARI) remain essentially unchanged.
> We report below the wall-clock times for fixed and random $\alpha$ in the first simulation study. The same scalability trends are observed.
>
> Wall-clock sec $\alpha = 1$ (new)
> | Algorithm | 150   | 300   | 600   | 1500  | 3000  | 7500 | 12000 |
> |-----------|-------|-------|-------|-------|-------|-------|-------|
> | Slice     | 1.27  | 2.26  | 4.24  | 10.10 | 20.05 | 50.42 | 89.42 |
> | Slice noa | 1.45  | 2.71  | 5.11  | 12.39 | 24.54 | 62.45 | na    |
> | BGS 10    | 1.14  | 2.06  | 3.85  | 9.22  | 18.53 | 47.14 | 78.48 |
> | BGS n     | 2.57  | 7.61  | 24.51 | na    | na    | na    | na    |
> | CRP wat   | 2.08  | 4.38  | 8.31  | 20.37 | 43.25 | na    | na    |
> | CRP       | 3.17  | 6.90  | 13.54 | 34.42 | 71.53 | na    | na    |
>
> Wall-clock sec  $\alpha$ random (Figure 2)
> | Algorithm | 150   | 300   | 600   | 1500  | 3000  | 7500 | 12000 |
> |-----------|-------|-------|-------|-------|-------|-------|-------|
> | Slice     | 1.31  | 2.21  | 4.22  | 10.14 | 19.96 | 51.06 | 86.59 |
> | Slice noa | 1.40  | 2.68  | 5.15  | 12.50 | 24.63 | 62.46 | na    |
> | BGS 10    | 1.13  | 2.04  | 3.97  | 9.34  | 18.53 | 47.97 | 79.60 |
> | BGS n     | 2.49  | 7.36  | 24.69 | na    | na    | na    | na    |
> | CRP wat   | 1.85  | 3.63  | 7.05  | 17.08 | 32.54 | na    | na    |
> | CRP       | 2.68  | 5.43  | 10.28 | 25.76 | 50.99 | na    | na    |
>
> - Mixing time analysis.
>
> Finally, for completeness, we would like to confirm that our theoretical results do not concern mixing times; however, this aspect is considered empirically in our numerical studies via the effective sample size (ESS) per second.

---

> > ### Author Rebuttal · Reviewer_U5pP · 2026-04-04
> >
> > I will maintain my rating and am happy to recommend acceptance.

---

> > > ### Author Response · Authors · 2026-04-07
> > >
> > > Thank you for your careful review and insightful suggestions. We greatly appreciate the positive recommendation for acceptance.

---

### Decision · Program_Chairs · 2026-04-30

**Decision:**

Accept (regular)

**Comment:**

This paper addresses a long-standing open problem in Bayesian nonparametrics by providing the first high-probability complexity guarantees for slice sampling in Dirichlet Process (DP) models. The authors establish that the dynamic truncation level, that is the number of components sampled beyond those occupied by data, is at most logarithmic in the sample size. This theoretical result is particularly significant because the computational cost of slice sampling was previously considered uncertain due to its random and potentially unbounded nature. By identifying the "singleton partition" as the worst-case scenario and proving uniformity across posterior cluster-growth regimes, this paper provides a rigorous foundation for the empirical scalability of these widely used exact inference approaches.

Despite some concerns regarding the significance of DP-based models in the broader machine learning landscape, which I partially share, my view is that the technical developments and foundational value of this paper justify its acceptance. The authors addressed some concerns regarding the empirical evaluation through the rebuttal, demonstrating that their results hold even when the DP concentration parameter is fixed rather than random. Although this study does not include a mixing time analysis, its per-iteration complexity bounds provide a quantitative tool for choosing exact slice samplers instead of biased truncation methods in large-scale applications. Overall, all reviewers expressed a positive opinion of this paper, and I'm therefore inclined to recommend that the paper is accepted.